# LLM Coaching LLM in Self-Play Training

## Abstract

Large language models (LLMs) have achieved impressive progress in domains such as mathematics and programming, supported by verifiable outputs and robust benchmarks. However, their potential in games—longstanding benchmarks for reinforcement learning (RL) research—remains underexplored. The most prominent challenge is no ground truth–high variance, especially in complex and strategic games like Texas Hold'em, where naively optimizing for the highest observed payoff risks training dead loops, while payoff estimation itself is highly resource-intensive. To address these issues, we propose the LLM Coach, which transforms raw self-play (SP) payoffs into class-wise reward functions by leveraging payoff data, state information, and the current policy. This design stabilizes training and accelerates learning. Within an RL+SP framework, our Qwen2.5-32B agent significantly outperforms strong baselines (e.g., Grok4, GPT-o3) in Texas Hold'em Poker, while also exhibiting improvements in broader capabilities.

## 1 Introduction

Large Language Models (LLMs) have demonstrated strong performance in Natural Language Processing (NLP). Reinforcement Learning (RL) with verifiable rewards can further enhance their capabilities in mathematics and programming Ouyang et al. (2022); Li et al. (2022); Shao et al. (2024), bringing them closer to human expert levels. However, current approaches face two key constraints: the scarcity of high-quality tasks with verifiable rewards, and the reliance on fixed RL environments (e.g., problem sets), which limits continual improvement. Game-playing tasks naturally address these issues. Games such as Go, Chess, and Texas Hold'em Poker are as complex as mathematics and programming while offering significantly greater diversity. Moreover, Self-Play Reinforcement Learning (SPRL) inherently enables continual evolution: by repeatedly competing against historical policies, an agent can in principle converge to the Nash Equilibrium (NE) policy. The success of AlphaGo Silver et al. (2016; 2018) exemplifies this paradigm, achieving superhuman performance without any prior human knowledge.

Despite the advantages of game tasks for LLMs in RL, previous work combining LLMs with Self-Play (SP) methods has failed to tackle the core challenge: the **no ground truth–high variance problem**. Unlike Supervised Fine-Tuning (SFT) or conventional LLM-based RL Guo et al. (2025), which directly rely on data or precise reward functions, SP lacks a ground truth baseline or direct reward function; instead, the reward must be inferred by competing against the agent's own historical policies. And it's important to note that LLM response (policy) reward and game payoff are not directly equivalent: naively optimizing a policy for maximum payoff can drive policies into dead loops—this is precisely what we refer to as the no ground truth problem. The high variance issue is more straightforward: in complex games such as Texas Hold'em Poker, payoff estimates exhibit extremely high variance, requiring millions of rollouts for reliable estimation. This inefficiency results in deep-network poker AIs being less powerful than table-based methods, which perform rollouts more efficiently Moravčík et al. (2017); Brown & Sandholm (2018; 2019).

In contrast, human experts require far fewer training samples. We argue that the key lies in their ability to "summarizing and learning by classification." In games, experts cluster similar states and update policies at the class level. These highly refined categories and their corresponding strategic maxims allow humans to avoid most incorrect actions in the initial stages of learning. However, past game AI designs were often tailored to specific games and completely lacked language and logical analysis capabilities. In contrast, modern LLMs already possess a basic understanding of game rules through pre-training and can conduct analysis and logical reasoning on game states via

Chain of Thought (CoT) Wei et al. (2022). Theoretically, LLMs can achieve the human-like effect of "summarizing and learning by classification."

The objective of this paper is to address the no ground truth-high variance problem in SPRL for LLMs. We first build an LLM-specific game environment using DeepMind OpenSpiel Lanctot et al. (2019), augmented with an Elo rating system Elo (1978) for quantitative evaluation. For training, we adopt the CFVFP Ju et al. (2024) algorithm to decompose sequential games into independent subgames, enabling direct integration with LLM RL methods such as Generative Reference Policy Optimization (GRPO) Shao et al. (2024) while preserving convergence guarantees to NE. Most critically, we introduce an LLM Coach into the architecture. Batched training data are processed by the Coach, which classifies states and proposes class-level policy updates.

Experimental results indicate that our method, built on Qwen2.5-32B and trained without GTO policies or SFT, surpasses strong baselines such as o3 and Grok4 in Texas Hold'em Poker after seven self-play iterations. The agent also demonstrates generalization to other games, while preserving overall capability without significant degradation, confirming the robustness and extensibility of our framework. Our self-play reinforcement learning environment and the trained models will be released as open source in the near future.

## 2 SELF-PLAY PRELIMINARIES FOR GAMES

The focus of this section is to introduce knowledge related to SP for learning in games. For the part concerning LLM post-training, please refer to App. A.

### 2.1 NORMAL-FORM GAMES, NASH EQUILIBRIUM AND FICTITIOUS PLAY

A Normal-Form Game (NFG) is the most fundamental model in game theory, used to describe static games. Define $\mathcal{N} = \{1, 2, \ldots, i, \ldots\}$ as the set of players, where each player $i$ has a finite action set $\mathcal{A}^i = \{a_1, a_2, \ldots\}$. The policy $\sigma^i$ of player $i$ is defined as a probability distribution over its action set $\mathcal{A}^i$. It should be noted that in this paper, the **policy** and **action** are strictly distinguished. Let $\Sigma^i$ denote the policy set of player $i$, where $\sigma^i \in \Sigma^i$. The policy profile $\sigma = \times_{i \in \mathcal{N}} \sigma^i$ represents the collection of all players' policies, and $\sigma^{-i}$ denotes the policy profile of all players except player $i$ within $\sigma$. The payoff is defined as $u^i : \Sigma \to \mathbb{R}$. It is also important to distinguish game payoff from RL reward, we use **payoff** instead of **reward** throughout this paper in games.

Solving a game problem is largely equivalent to finding the Nash Equilibrium (NE) of the game. As a core concept in game theory, NE was proposed by John Nash in the 1950s Nash (1951). Its significance lies in the fact that under NE, each player's policy is the BR to the policies of other players, no individual has an incentive to unilaterally change their policy. We first define the Best Response (BR) policy. Formally, given the opponents' policy $\sigma^{-i}$, the BR policy of player $i$ is defined as:

$$b^i(\sigma^{-i}) = \arg\max_{a \in \mathcal{A}^i} u^i(a, \sigma^{-i}). \tag{1}$$

Based on the BR policy, the exploitability $\epsilon^i$ of player $i$ with respect to the opponents' policy $\sigma^{-i}$ is defined as:

$$\epsilon^i = u^i(b^i(\sigma^{-i}), \sigma^{-i}) - u^i(\sigma), \tag{2}$$

the total exploitability of the policy profile is $\epsilon = \sum_{i \in \mathcal{N}} \epsilon^i$. When $\epsilon = 0$, it means that no player can gain any benefit by deviating from the current policy, i.e., this policy profile constitutes a NE.

Finding the NE of a game is a PPAD problem Daskalakis et al. (2009). Against this backdrop, recent studies have shifted their focus to solving NE through iterative self-play method Zinkevich et al. (2007); Lanctot et al. (2009); Heinrich et al. (2015). Fictitious Play (FP) Robinson (1951) is one of the earliest classic self-play methods. In FP, let the initial average policy profile be $\bar{\sigma}_{t=1}$. In each round, player $i$ assumes that the opponents will use the average policy from previous rounds, $\bar{\sigma}_t^{-i}$, calculates the BR policy for the current round, $b^i(\bar{\sigma}_t^{-i})$, and updates its own policy $\sigma^i$ in the direction of this BR. The update formula is as follows:

$$\bar{\sigma}_{t+1} = (1 - \alpha_{t+1})\bar{\sigma}_t + \alpha_{t+1}b(\bar{\sigma}_t), \tag{3}$$

where $\alpha_t$ is the step-size factor. In original FP $\alpha_t = 1/(t+1)$. As $t \to \infty$, $\bar{\sigma}_{t+1}$ converges to the NE in two-player zero-sum games.

## 2.2 EXTENSIVE-FORM GAMES AND COUNTERFACTUAL VALUE BASED FICTITIOUS PLAY

Games such as Go, Chess, and Texas Hold'em are typically represented using Extensive-Form Games (EFG), which are used to characterize sequential games. Each state in the game tree is represented by a node $s \in \mathcal{S}$, where terminal states constitute the set of leaf nodes $\mathcal{Z} \subset \mathcal{S}$. For each non-terminal state $s \in \mathcal{S} \setminus \mathcal{Z}$, the action set $\mathcal{A}(s)$ represents all available actions, corresponding to the edges that lead to the next node. The player function $P : \mathcal{S} \rightarrow \mathcal{N} \cup \{c\}$ specifies the player at each node, where $c$ denotes a chance node. In games with incomplete information, the set of states that a player $i$ cannot distinguish is grouped into an information set $I \in \mathcal{I}^i$. The payoff function $R : \mathcal{Z} \rightarrow \mathbb{R}^{|\mathcal{N}|}$ maps each terminal state to a payoff vector for all players. A behavioral policy $\sigma(I) \in \mathbb{R}^{|\mathcal{A}(I)|}$ defines the probability distribution of player $i$ over each available action at the information set $I$.

Counterfactual Value Based Fictitious Play (CFVFP) is a variant of FP applied to EFGs. The counterfactual $Q$-value is defined as:

$$Q_t^i(I, a) = h_{\bar{\sigma}_t}^{-i}(I)u^i\left(I, \bar{\sigma}_t|_{I \rightarrow a}\right), \forall I \in \mathcal{I}. \tag{4}$$

Here, $h_{\sigma_t}^{-i}(I)$ represents the occurrence probability of the information set $I$ when all players (excluding $i$, including chance nodes) select actions according to $\bar{\sigma}_t$. $u(I, \sigma)$ is defined as the expected payoff when all players follow the policy $\sigma$ at the information set $I$. Let $\sigma|_{I \rightarrow a}$ denote the policy profile where the policy at the information set $I$ is $a$, while the policies at all other information sets remain the same as in $\sigma$.

Similar to FP, CFVFP solves for the BR policy based on behavioral policies. The BR policy at the information set $I$ at time $t + 1$ is:

$$\sigma_{t+1}^i(I) = \underset{a \in \mathcal{A}^i(I)}{\arg\max} Q_t^i(I, a), \forall I \in \mathcal{I}, \tag{5}$$

after $T$ iterations, the average policy $\bar{\sigma}_T^i$ at the information set $I$ is:

$$\bar{\sigma}_T^i(I) = \frac{\sum_{t=1}^{T} h_{\sigma_t}^i(I)\sigma_t^i(I)}{\sum_{t=1}^{T} h_{\sigma_t}^i(I)}. \tag{6}$$

Eventually, $\bar{\sigma}_T$ will converge to the NE as $T \rightarrow \infty$.

## 3 MOTIVATION

Classical SPRL algorithms such as CFVFP and Deep CFR Brown et al. (2019); Steinberger et al. (2020), as well as advanced AIs like AlphaGo and DeepStack Moravčík et al. (2017), face a fundamental bottleneck: the absence of ground truth and the high variance of payoff estimates. In LLM-based SP, we therefore focus on two design axes: how to construct a surrogate "ground truth" suitable for LLM RL and how to reduce the variance of RL reward estimation.

### 3.1 GROUND TRUTH CONSTRUCTION IN LLM RL FOR GAMES

Several studies have combined SP with LLMs Xu et al. (2023); Kuba et al. (2025); Liu et al. (2025), but most construct RL rewards simply by maximizing action payoffs, which may leads to cycles in long-term SP training Bowling & Veloso (2002). For example, cycles of the form $\sigma_A \succ \sigma_B$, $\sigma_B \succ \sigma_C$, and $\sigma_C \succ \sigma_A$ ($\succ$ means outperforms) can arise and preclude convergence to an NE.

First, we elicit policy-level outputs rather than single actions. For example, in Rock–Paper–Scissors (RPS), prior work typically asks "What action do you choose?", whereas we ask "What is your probability distribution over Rock, Paper, and Scissors?". This shift offers three advantages: (i) since an NE is defined over policies, directly outputting policies better adapts to equilibrium learning algorithms; (ii) in games like Texas Hold'em, humans similarly first determine policy and then select actions based on this; and (iii) policy-level outputs mitigate LLM biases. Although LLMs may recognize that the equilibrium in RPS is $[1/3, 1/3, 1/3]$, they often overproduce "Rock" Xu et al. (2023); Vidler & Walsh (2025); Bailis et al. (2024). We argue that this bias stems not from

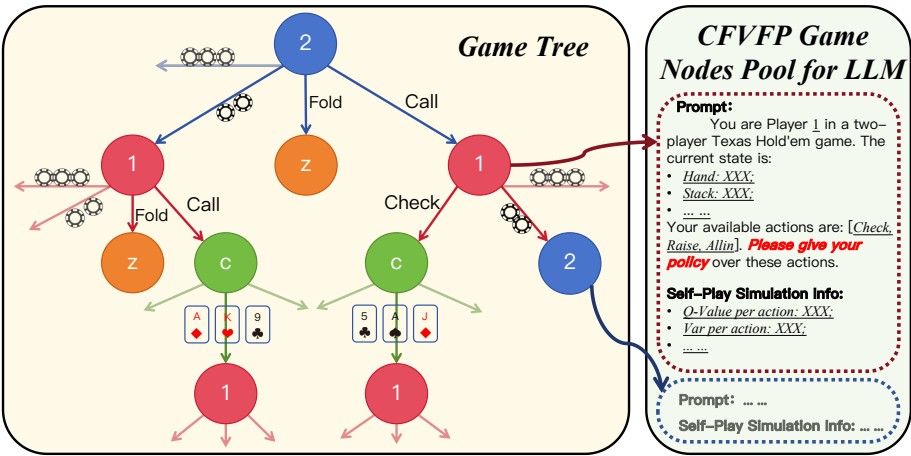

Figure 1: Illustration of the data set in SPRL framework. Game states and simulation outcomes are collected into a CFVFP game node pool and converted into structured prompts for the LLM. The LLM then outputs policy distributions over available actions, which are compared with counterfactual Q-values derived from SP simulations. This conversion makes the game dataset consistent in form with common mathematics or programming datasets.

poor strategic ability, but rather from the fact that prior arena environment are always static, which encourages impulsive single-action selection. By prompting for full policy distributions, this issue is eliminated.

Secondly, conventional LLM RL operates on state–response pairs, which is incompatible with sequential games. Two approaches exist: redesign LLM-based RL frameworks for sequential decision-making, or decompose games into independent nodes. While the former is theoretically valuable, it is impractical in the short term. We therefore adopt the latter. Specifically, following CFVFP, we decompose the game tree and optimize LLM policies (response) at the node level, as illustrated in Fig. 1. For each information set, simulated rollouts provide counterfactual Q-value estimates as shown in Eq. 4, which are then used to derive an ideal reference policy, such as those in Eq. 5 and Eq. 6. During RL, the LLM-generated policy is compared with this reference, and higher similarity yields higher reward. This formulation transforms SP into a standard RL problem, while ensuring stable convergence to an NE.

## 3.2 LLM Coach

In games such as Texas Hold'em, which exhibit very high variance, extremely large-scale simulations are required to obtain reliable signals. By contrast, human experts require significantly less data. The difference stems from their update logic. RL agents typically conduct single-state optimization, which fails to generalize effectively across related scenarios directly. However, human players cluster similar positions, summarization general rules (e.g., hand strength ranking in poker or endgame motifs in Chess), and update policies at the class level. This mechanism enables them to converge with limited data.

Early deep RL architectures based on CNNs or long LSTMs lack the capacity for high-level abstraction, as they cannot interpret or generate language. Consequently, they rely on engineered domain knowledge, such as manually crafted poker hand abstractions, to reach comparable efficiency. LLMs provide a natural alternative. Through pre-training, they implicitly acquire elements of game rules and basic strategies, and—via chain-of-thought reasoning—can symbolically describe and summarize game states. Leveraging these strengths, we introduce the **LLM Coach** (Fig. 2). Rather than applying evaluation results directly to policy updates, we form tuples of the state, the original LLM policy, and the self-play evaluation output, which are then processed by the LLM Coach. The Coach clusters states, diagnoses weaknesses in the original policy, and produces class-level refinements.

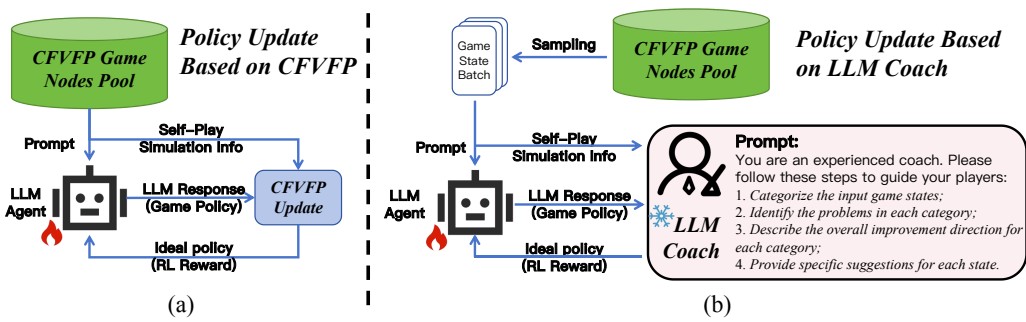

Figure 2: (a) Policy update based on CFVFP, where SP simulations provide counterfactual Q-values for direct policy improvement by Eq. 10; (b) Policy update based on LLM Coach, where sampled game states and policies are classified, analyzed, and improved via language-guided reasoning to produce updated policies.

This mechanism enables higher-level optimization, reduces the sample demand of self-play, and narrows the efficiency gap between AI agents and human experts.

## 4 LLM GAME ENVIRONMENT AND EVALUATION

Prior to training LLMs with games, it is necessary to construct a foundational framework that enables autonomous game interactions. Building on DeepMind's OpenSpiel Lanctot et al. (2019), this study develops a customized framework tailored for LLMs. In contrast to OpenSpiel, it adds three key functions: textual conversion of game states and actions, verification of action validity, and parallel feedback of game outcomes. Using this framework, we further design a quantitative evaluation system for LLMs' game-theoretic capabilities through a scenario-adapted approach. For games with binary outcomes (e.g., Chess), the Elo rating system Elo (1978) is used to calculate the LLM's skill level. For games with continuous payoff distributions (e.g., Texas Hold'em), Brownian motion diffusion approximations convert payoff values and variances into standardized win-rate metrics. These metrics are then evaluated with the Elo system to compare model performance. Technical details of the framework and evaluation system are provided in App. B.

## 5 METHOD

### 5.1 POLICY-BASED LLM SPRL FRAMEWORK

Based on the CFVFP algorithm, we construct a LLM improvement method under the SPRL framework. Let $\theta$ denote the parameters of the LLM, with $\theta_t$ representing the parameters at the $t$-th stage. The behavioral policy of the LLM with parameters $\theta_t$ at information set $I$ is denoted as $\omega_{\theta_t}(I)$. According to CFVFP, the counterfactual $Q$-value of each action at $I$ is given by

$$Q^i_{\theta_t}(I, a) = h^{-i}_{\omega_{\theta_t}}(I)u^i\left(I, \omega_{\theta_t}|_{I \to a}\right), \forall I \in \mathcal{I}. \tag{7}$$

In large-scale games, exact computation of $h^{-i}_{\omega_{\theta_t}}(I)$ and $u^i\left(I, \omega_{\theta_t}|_{I \to a}\right)$ for all $I$ is infeasible. To address this issue, we allow $\omega_{\theta_t}$ to act freely, then collect sampled information sets $I \sim \mathcal{I}$ into a data pool $\mathcal{D}_{\text{sample}} \subseteq \mathcal{I}$. The estimated counterfactual $Q$-value is defined as:

$$\hat{Q}^i_{\theta_t}(I, a) = \hat{u}^i_k\left(I, \omega_{\theta_t}|_{I \to a}\right), \forall I \in \mathcal{D}_{\text{sample}}, \tag{8}$$

where $\hat{u}^i_k$ is obtained by $k$ simulations from $I$ under $\omega_{\theta_t}$, $\hat{Q}^i_{\theta_t}(I, a)$ is an unbiased estimator of $Q^i_{\theta_t}(I, a)$. $\forall I \in \mathcal{D}_{\text{sample}}$, the BR policy $\sigma^{i,*}(I)$ is:

$$\sigma^{i,*}(I) = \underset{a \in \mathcal{A}(I)}{\arg\max}\hat{Q}^i_{\theta_t}(I, a). \tag{9}$$

The CFVFP approach defines the ideal policy as:

$$\omega^*(I) = (1 - \alpha)\omega_{\theta_t}(I) + \alpha\sigma^{i,*}(I), \tag{10}$$

in standard CFVFP $\alpha = \frac{1}{t+1}$. For LLM updates, this choice induces instability in early stages and noise dominance later. We fix $\alpha = 0.1$ in this work. The updated parameters $\theta^*$ satisfy $\omega^*(I) \leftarrow \omega_{\theta^*}(I)$. We then construct a CFVFP Node dataset (analogous to a mathematics dataset) as follows:

$$\mathcal{D}_{\text{CFVFP}} = \{(I, \omega^*(I)) \mid \forall I \in \mathcal{D}_{\text{sample}}\}, \tag{11}$$

and design a RL reward that measures the distance between $\omega_\theta(I)$ and the ideal policy $\sigma^{i,*}(I)$. A smaller distance yields a higher reward. Using weighted absolute differences, we define:

$$R(\omega_\theta, I, \omega^*(I)) = \sum_{a \in \mathcal{A}(I)} w_a \left|\omega_\theta(I, a) - \omega^*(I, a)\right|. \tag{12}$$

Numerical settings of $w_a$ are provided in App. C.

## 5.2 LLM COACH

The CFVFP-based SPRL algorithm yields the update rule Eq. 10. However, replacing $\omega_{\theta_t}$ with table storage or deep networks does not fundamentally alter convergence. These approaches address the absence of ground truth but fail to reduce high variance.

Traditional poker AIs (e.g., Pluribus, Cepheus, DeepStack) provide useful insights, mainly in two aspects. First, *state abstraction*: instead of handling individual information sets, they classify hands into strength categories (e.g., the "9 Bucket" method partitions 169 hand combinations into 9 levels). Second, *belief updating*: during policy updates, they simultaneously update the full opponent hand distribution, such as the $C_{52}^2 = 1326$ pre-flop combinations. The core idea of the aforementioned traditional algorithms can be summarized as "summarizing and learning by classification". When humans engage in games, there are no explicit formulas for information clustering or opponent hand inference; such decisions are completed through natural language reasoning in the brain.

Motivated by this, we introduce *LLM Coach* ($\theta_{\text{coach}}$), replacing direct policy updates with a reflective process. We collect counterfactual values $Q_{\theta_t}(I)$, confidence intervals, information sets $I$, and prior policies $\omega_{\theta_t}(I)$ into a pool:

$$\mathcal{D}_{\text{coach input}} = \{(Q_{\theta_t}(I_1), I_1, \omega_{\theta_t}(I_1)), (Q_{\theta_t}(I_2), I_2, \omega_{\theta_t}(I_2)), \cdots\}. \tag{13}$$

This pool is fed into $\theta_{\text{coach}}$, whose prompt specifies three tasks: classify information sets, analyze flaws of $\omega_{\theta_t}(I)$ per class, and output improved policies $\omega^*_{\text{coach}}(I)$. The result is

$$\mathcal{D}_{\text{coach}} = \{(I_1, \omega^*_{\text{coach}}(I_1)), (I_2, \omega^*_{\text{coach}}(I_2)), \cdots\}. \tag{14}$$

At last, the RL reward for the LLM policy $\omega_\theta$ on $(I, \omega^*(I)) \sim \mathcal{D}_{\text{coach}}$ is:

$$R(\omega_\theta, I, \omega^*(I)) = -\sum_{a \in \mathcal{A}(I)} w_a \left|\omega_\theta(I, a) - \omega^*(I, a)\right|. \tag{15}$$

This design bypasses explicit abstraction and belief-updating steps, enabling policy improvement in a way closer to human reasoning. For reflective prompt design, see App. D.

# 6 EXPERIMENTS

## 6.1 SETUP

For all experiments, we adopt Qwen2.5-32B Yang et al. (2024) as the base model. Our training framework builds on Verl Sheng et al. (2024) framework, the reinforcement learning algorithm is GRPO, and the optimizer is Adam Kingma (2014). We conduct seven self-play iterations (denoted as SPRL-1 to SPRL-7). Each iteration consists of 50 training steps, with 128 samples per step, corresponding to 6,400 CFVFP nodes. Since the LLM coach cannot guarantee valid outputs for all CFVFP nodes, we collect 8,000 CFVFP nodes per iteration during the data collection stage. For hardware resource, the training phase is conducted on 16 servers equipped with 16 H20 GPUs each (256 GPUs in total) for approximately 30 hours, while the CFVFP node collection phase is performed on 16 servers equipped with 8 H20 GPUs each (128 GPUs in total) for about 20 hours.

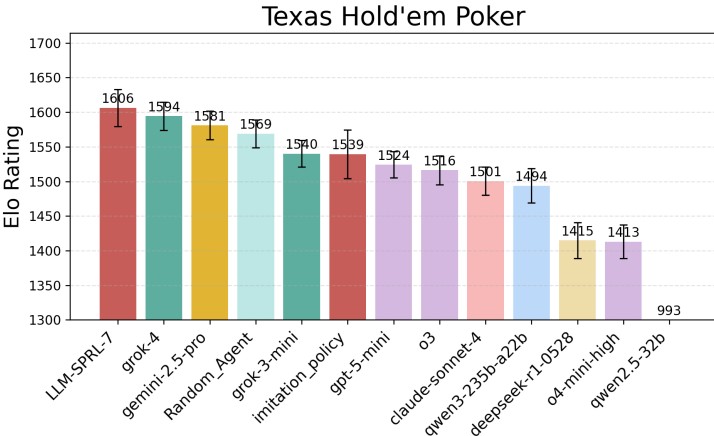

Figure 3: Elo ratings of different models in Texas Hold'em Poker. Our model (in red) achieves the highest score. Error bars indicate 95% confidence intervals.

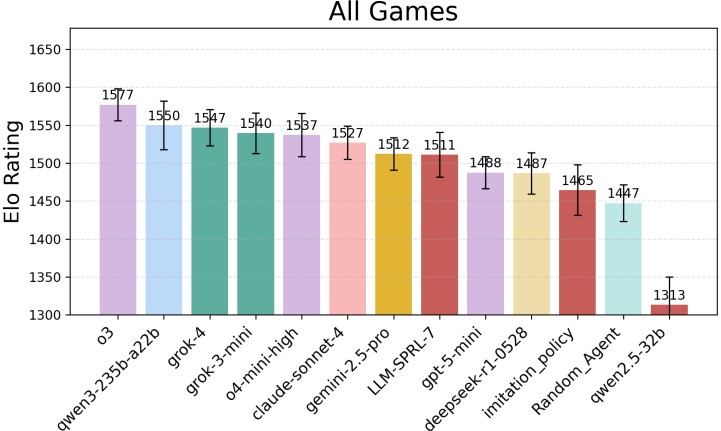

Figure 4: Elo ratings of different models across six games (Texas Hold'em, Chess, and others). Although our model is trained solely on Texas Hold'em, it demonstrates generalization by achieving competitive improvements in the aggregated evaluation. Error bars represent 95% confidence intervals.

## 6.2 IMITATION LEARNING

This study adopts Qwen2.5-32B as the base model. However, the raw Qwen2.5-32B performs extremely poorly in Texas Hold'em: it often fails to generate valid policies, achieves only an Elo score of 998, and records an average win rate below 5%, far behind the other models. To obtain a usable starting point, we introduce an imitation learning phase. Specifically, ChatGPT-5-mini and Grok-3-mini engaged in free play, yielding 28,500 valid samples (21,000 from ChatGPT-5-mini and 7,500 from Grok-3-mini). Using this dataset as the imitation target, Qwen2.5-32B was trained to approximate the output policy of the two models. After imitation learning, the model acquired the basic ability to output reasonable policies and reached an Elo score of 1539, close to Grok-3-mini's 1540 as shown in Fig. 3. Nevertheless, this performance still falls significantly short of stronger models such as GPT-o3 and Grok-4, indicating that imitation learning merely provides an initial capability rather than closing the competitive gap.

## 6.3 LLM COACH

The performance of the LLM coach can be assessed from three perspectives. First, we examine macro-level policy consistency. Although the ideal policies derived from traditional CFVFP, denoted as $\omega^*_{\text{CFVFP}}$, are subject to stochastic noise, they should remain largely consistent with those produced by the LLM Coach, $\omega^*_{\text{LLM Coach}}$. To verify this, we computed the absolute distance between the two policies during the SP-3 to SP-4 stage: $\mathbb{E}_{a \in \mathcal{A}(I), I \in \mathcal{D}_{\text{SP-3}}} |\omega^*_{\text{CFVFP}}(I, a) - \omega^*_{\text{LLM Coach}}(I, a)| \leq 0.03$ (In Texas Hold'em, this kind of error is negligible), confirming their expected alignment.

Second, we examine the correctness of situation analysis. Since the LLM Coach applies structured reasoning via CoT, its qualitative outputs can be inspected. As illustrated in App. D.2, the Coach performs class-wise updates: even when an action's estimated payoff is positive, its probability may be reduced if similar states suggest weaker long-term returns. This indicates an ability to incorporate broader contextual factors beyond single-state optimization.

|  | **LLM Coach** | **CFVFP** |
|---|---|---|
| Elo Score | $1549 \pm 32$ | $1470 \pm 31$ |

Table 1: Comparison of Elo scores obtained with LLM Coach versus CFVFP guidance in SP-3.

Finally, we compare RL training outcomes based on policies generated by the two approaches. In SP-3, both were distilled into the LLM. As shown in Tab. 1, the resulting policies reached Elo scores of 1549 (LLM Coach) and 1470 (CFVFP). This significant margin demonstrates the superior effectiveness of the LLM Coach and underscores that, under limited rollouts (6400), traditional algorithms struggle to yield accurate ideal policies, whereas the Coach provides a more reliable alternative.

## 6.4 SELF-PLAY

After imitation learning, the self-play process begins. Opposing policies from previous stages are simulated, interaction data are collected, and the data are passed to the LLM Coach to generate improved reference policies. RL is then performed using the distance between the LLM's policy and the reference policy as the reward. Ideally, the model $\theta_{t+1}$ should be trained on top of $\theta_t$. In practice, however, LLM RL causes a rapid entropy collapse, which restricts the feasible number of fine-tuning iterations. Consequently, all fine-tuning procedures in this study are initialized from the original Qwen2.5-32B model. As shown in Fig. 3, after 7 rounds of SP, our model surpasses models such as Grok4 and o3 in Texas Hold'em. For the capabilities of the remaining SP models, please refer to Fig. 5.

## 6.5 GENERALIZATION ABILITY OF THE MODEL

We evaluate the model's generalization from two perspectives. First, as shown in Fig. 4 we test six game-theoretic tasks—Texas Hold'em, Chess, Liars Dice, Goofspiel, Leduc Poker Shi & Littman (2002), and Taboo Words—covering diverse difficulty levels and game categories . Second, we assess six broader capabilities: mathematics (AIMI 2025, Olympiad Bench He et al. (2024a)), coding (LiveCode Bench Jain et al. (2024)), instruction (IFEval Zhou et al. (2023)), knowledge (Chinese SimpleQA He et al. (2024b)), reasoning (ZebraLogic Bench Dziri et al. (2024)), and overall performance (MMLU Hendrycks et al. (2020)).

Across the game-theoretic tasks, the model exhibits improvements in all cases except Taboo Words. The most substantial gain is observed in Leduc Poker (Fig. 8), which closely resembles Texas Hold'em, where the model's Elo ranking advances from last to second place. In contrast, for complete-information games such as Chess (Fig. 11), the improvement is marginal, with performance only slightly exceeding that of a random policy. For Taboo Words, which lacks a well-defined game-tree structure, training not only fails to improve performance but even leads to degradation, a result consistent with intuition. These results can be found in App. E.

Across general capabilities in Tab. 2, the model shows slight gains in instruction following but small declines elsewhere. This likely reflects the strict action constraints in our setup, which indirectly

| Model | AIME 2025 | Olympiad Bench | LiveCode Bench | IFEval | Chinese SimpleQA | Zebra LogicBench | MMLU |
|---|---|---|---|---|---|---|---|
| SP-7 (ours) | 47.8 | 81.3 | 46.6 | 58.4 | 32.9 | 64.0 | 80.0 |
| Imitation (ours) | 41.1 | 78.4 | 47.3 | _58.7_ | 33.1 | 66.5 | 76.9 |
| Qwen2.5-32B | _50.0_ | _81.6_ | _49.6_ | 57.9 | _34.0_ | _67.5_ | _81.3_ |
| DeepSeek-V3-0324 | 44.0 | 72.0 | 40.5 | 86.3 | 71.4 | 83.4 | 89.3 |
| OpenAI-gpt4.1-0414 | 34.7 | 65.8 | 35.9 | 91.1 | 62.0 | 52.5 | 89.9 |

Table 2: Evaluation of different models across seven datasets.

sharpen instruction following. Since no extra general-purpose data were used, modest regressions are expected. Prior work Liu et al. (2025) suggests game-play training can enhance mathematical and reasoning skills, but effects depend on game scale. In toy games (e.g., Kuhn Poker Kuhn (1950)), LLMs often compute odds or win rates, improving mathematical ability. In Texas Hold'em, however, decisions rely mainly on language reasoning, yielding no math gains. Sample LLM outputs in Texas Hold'em is shown in App. B.1.1.

## 7 RELATE WORK

Recent studies have explored combining LLMs with self-play, though with distinct goals. Language Self-Play (LSP) Kuba et al. (2025) formulates question generation and answering as an adversarial game, with the primary aim of producing higher-quality data for bootstrapped training rather than improving game-playing competence. Liao et al. (2025) focus on bridging the gap between "what" and "how" in reasoning, relying exclusively on human expert data for RL without employing genuine self-play. Cicero Bakhtin et al. (2022) combines dialogue modeling with strategic planning in Diplomacy, but its performance gains are driven primarily by traditional game-learning algorithms (e.g. Zhang et al. (2022)), rather than by direct LLM-level self-play. In the Werewolf setting Bailis et al. (2024), integrate an RL selector with LLM reasoning to alleviate output biases; the emphasis is on strengthening logical coherence and linguistic expressiveness rather than computing equilibria. Similarly, the Self-Playing Adversarial Game (SPAG) Cheng et al. (2024) employs reward signals derived from self-play in a taboo-style adversarial task to enhance reasoning ability, with its objective centered on inference quality rather than equilibrium learning.

Among works targeting game performance, most pursue gains against fixed opponents rather than convergence to a NE. Divide-Fuse-Conquer (DFC) Zhang et al. (2025) employs grouping, parameter fusion, and stability heuristics to improve robustness and diversity, but not NE computation. Wang et al. (2025) demonstrate improved poker performance, yet still without addressing NE directly.

The research most closely related to ours is SPIRAL Liu et al. (2025), which applies self-play training in Kuhn Poker, Tic-Tac-Toe, and simple negotiation tasks, and observes that stronger gameplay contributes to LLM reasoning skills. However, from the perspective of game-theoretic complexity (e.g., the scale of state and action spaces), the games considered by SPIRAL remain relatively simple; consequently, its variance-reduction mechanisms largely follow conventional designs and do not address efficient training at the policy-level.

## 8 CONCLUSIONS AND FUTURE WORK

This paper introduces an LLM-based SPRL framework that integrates CFVFP with an LLM Coach, addressing two long-standing challenges: lack of ground truth and high variance. By eliciting policy-level outputs, decomposing sequential games into node-level updates, and leveraging the Coach for state clustering and class-level refinements, the framework improves both learning stability and efficiency. Experiments in Texas Hold'em Poker show that Qwen2.5-32B surpasses strong baselines such as Grok-4 and GPT-o3, while also achieving gains across diverse games. Future work will extend the framework to a broader set of games and develop mechanisms for the coordinated evolution of both gamer and coach models. Balanced co-adaptation is expected to sustain progress and may offer a promising path toward enhancing AGI in complex strategic environments.

ETHICS STATEMENT

The research presented does not involve human subjects, personally identifiable information, or sensitive data, and it does not introduce foreseeable risks of misuse. The study is conducted solely on publicly available benchmarks and standard experimental environments. To the best of our knowledge, there are no ethical concerns associated with the methods, data, or results reported in this paper.

REPRODUCIBILITY STATEMENT

We have taken several steps to ensure the reproducibility of our results. The paper and appendix provide detailed descriptions of the algorithms, training procedures, evaluation protocols, and datasets used. In addition, we will release the training environment configuration and the final trained models after the review process to facilitate independent verification. Together, these resources will enable other researchers to reproduce and extend our findings.

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

## A  REINFORCEMENT LEARNING OF LLMS

In this paper, an LLM parameterized by $\theta$ is defined as a policy $\pi_\theta$. Let $x$ denote a query, and $\mathcal{D}$ denote the query set. For a response $y$ to query $x$, its likelihood under policy $\pi_\theta$ is expressed as $\pi_\theta(y \mid x) = \prod_{t=1}^{|y|} \pi_\theta(y_t \mid x, y_{<t})$, where $|y|$ represents the number of tokens in response $y$. A verifier $r$ can score the query-response pair $(x, y)$ to generate a reward $R(x, y) \in \mathbb{R}$.

### A.1  PROXIMAL POLICY OPTIMIZATION (PPO)

PPO (Schulman et al., 2017) leverages samples generated by the old policy $\pi_{\theta_{\text{old}}}$ and restricts policy updates within a proximal region of the old policy using a clipping mechanism. Specifically, PPO adopts the following objective function for policy optimization (for brevity, the KL regularization term is omitted below as it is not the focus of this paper):

$$\mathcal{J}_{\text{PPO}}(\theta) = \mathbb{E}_{x \sim \mathcal{D}, y \sim \pi_{\theta_{\text{old}}}(\cdot|x)} \left[ \frac{1}{|y|} \sum_{t=1}^{|y|} \min \left( w_t(\theta) \hat{A}_t, \text{clip} \left( w_t(\theta), 1 - \varepsilon, 1 + \varepsilon \right) \hat{A}_t \right) \right]$$

Here, the importance ratio for token $y_t$ is defined as $w_t(\theta) = \frac{\pi_\theta(y_t|x, y_{<t})}{\pi_{\theta_{\text{old}}}(y_t|x, y_{<t})}$, and the advantage value $\hat{A}_t$ of $y_t$ is estimated by a separate value model. $\varepsilon$ denotes the clipping range for the importance ratio.

In practical applications, the core challenge of PPO lies in its strong dependence on the value model. Specifically, the scale of the value model is usually comparable to that of the policy model, which imposes significant memory and computational burdens. Additionally, the effectiveness of the algorithm relies on the reliability of value estimates. However, obtaining a reliable value model is inherently challenging, and adapting it to longer responses and more complex tasks presents even greater difficulties.

### A.2  GROUP RELATIVE POLICY OPTIMIZATION (GRPO)

GRPO (Shao et al., 2024) eliminates the need for a value model by calculating the relative advantage value of each response within a group of responses generated for the same query. Specifically, GRPO employs the following objective function for optimization:

$$\mathcal{J}_{\text{GRPO}}(\theta) = \mathbb{E}_{x \sim \mathcal{D}, \{y_i\}_{i=1}^G \sim \pi_{\theta_{\text{old}}}(\cdot|x)} \left[ \frac{1}{G} \sum_{i=1}^{G} \frac{1}{|y_i|} \sum_{t=1}^{|y_i|} \min \left( w_{i,t}(\theta) \hat{A}_{i,t}, \text{clip} \left( w_{i,t}(\theta), 1 - \varepsilon, 1 + \varepsilon \right) \hat{A}_{i,t} \right) \right]$$

where $G$ represents the number of responses generated for each query $x$ (i.e., group size). The importance ratio $w_{i,t}(\theta)$ and advantage value $\hat{A}_{i,t}$ for token $y_{i,t}$ are defined respectively as:

$$w_{i,t}(\theta) = \frac{\pi_\theta(y_{i,t} \mid x, y_{i,<t})}{\pi_{\theta_{\text{old}}}(y_{i,t} \mid x, y_{i,<t})}, \quad \hat{A}_{i,t} = \hat{A}_i = \frac{r(x, y_i) - \text{mean}\left(\{r(x, y_i)\}_{i=1}^G\right)}{\text{std}\left(\{r(x, y_i)\}_{i=1}^G\right)}$$

It is important to note that all tokens within response $y_i$ share the same advantage value $\hat{A}_i$.

## B  DETAILS OF LLM GAME ENVIRONMENT AND EVALUATION

### B.1  LLM-BASED SELF-GAME FRAMEWORK

Compared with OpenSpiel, our framework incorporates three additional core functions: (i) textual conversion of game states and actions, (ii) an action validity verification mechanism, and (iii) parallel feedback of game outcomes. Specifically, textual conversion transforms non-textual information (e.g., states and actions) into natural language descriptions that can be directly interpreted by LLMs, serving as both inputs and outputs. The validity verification mechanism ensures that actions generated by LLMs comply with game rules. Parallel feedback then computes and returns

round-level outcomes (e.g., win/loss or payoff values) simultaneously. Benefiting from the modular design of OpenSpiel, our environment can be easily extended to new games by providing the corresponding game descriptions. To comprehensively evaluate model capabilities, we select a diverse set of games: Texas Hold'em (a high-difficulty, incomplete-information game), Chess (a high-difficulty, complete-information game), Liars Dice (a medium-difficulty, incomplete-information game), Goofspiel (a medium-difficulty, complete-information game), Leduc Poker Shi & Littman (2002) (a low-difficulty, incomplete-information game), and Taboo Words (a language-based game).

### B.1.1 TEXAS HOLD'EM

The game prompt is:

```
# Texas Hold'em Decision Prompt

You are a player in a **Texas Hold'em game with N players**.
Your player ID is **i** (starting from 0).

## Game Information
- Each player starts with an initial stack of **init_stack**
- Small blind = 1, Big blind = 2

## Available Actions
- '[Fold]': fold and give up the current hand
- '[Call]' : call the current bet
- '[Raise]' : raise on top of the current bet
- '[Allin]' : bet all remaining chips

**Notes:**
- In this simplified version, *Call* and *Check* are not
   distinguished.
- '[Raise]' is always fixed to 1.5 times the current pot.

## Output Format
- Your answer must be a probability distribution over the four
   actions '[Fold]', '[Call]', '[Raise]', '[Allin]'.
- Example: '[0.2, 0.3, 0.4, 0.1]'
- The probabilities must sum to 1.

## Legal Actions
- In some situations, not all four actions are legal (e.g., '[Fold
   ]' may be a dominated strategy).
- You will always be given the set of legal actions for the
   current round.
- Your probability distribution must **exactly match the legal
   actions** in order and still sum to 1.

## Current Game Context
- Public cards: [ Public ]
- Current pot: [ Pot ]
- Game sequence so far: [ Sequences ]
  - 'c' = call/check
  - 'rx' = raise to x (x is an integer)
- Your private cards: [ Private ]
- Your legal actions: [ Legal Actions ]
```

A typical LLM answer is:

```
<think>
```

```
Let me analyze this Texas Hold'em situation. First, Player 0 holds
    Qs5h, which does not look very strong; even making a pair may
    not be good enough. The community cards are still empty, and
    the pot is 4. The game history shows [c], meaning the previous
    player has already called.

The legal actions are Call, Raise, and All-in. I need to consider
    the probability of each option. Since the hand is mediocre,
    Call seems to be the safe choice. If raising is possible, I
    need to account for the opponent's potential responses. All-in
    is too risky and should only be used with a strong chance of
    winning. Therefore, the probability distribution should lean
    toward Call, followed by Raise, and finally All-in. The total
    must sum to 1 and match the order of the legal actions.
</think>
<action>
[0.7, 0.2, 0.1]
</action>
```

### B.1.2  CHESS

```
# Chess Decision Prompt

You are playing a game of **Chess** under standard Chess rules.
The game state follows the **OpenSpiel** format.

---

## Current Position
[ information_state_string ]

---

## Legal Actions
[ legal_actions_str ]

---

## Action Instruction
Choose your next move from the list of legal actions.
The format should follow standard algebraic notation, e.g., ''e3
    '', ''f4''.
```

### B.1.3  GOOFSPIEL

```
# Goofspiel (GOPS) Decision Prompt

You are playing a **two-player Goofspiel (GOPS, card bidding)**
    game.

---

## Game Rules
1. **Cards:**
    - **Prize deck:** 1-{num_cards} (a total of {num_cards} cards,
        valued 1-{num_cards}), placed face down.
```

```
    - **Player hands:** each player holds another suit with the
      same {num_cards} cards, visible only to themselves.

2. **Round Procedure:**
   1. Flip the top card of the prize deck -> this becomes the *
      prize card*.
   2. Both players secretly choose one card from their hand as
      their *bid*.
   3. Reveal both bids simultaneously and compare values:
      - Higher bid wins the prize card (points are added to the
        score).
      - **If tied -> Burn Rule:** the prize card and both bid
        cards are discarded, no points awarded.
   4. Regardless of outcome, both bid cards are discarded and
      cannot be reused.

3. The game lasts {num_cards} rounds. After all prize cards are
   revealed, the player with the higher total score wins; equal
   scores result in a draw.

---

## Action Instruction
Choose **1 card** from {legal_actions} as your bid.
Output only the chosen card value (no spaces, no explanations).
Examples: '2', '5'.

- Note: In the *Win sequence*, '-3' denotes a draw.

---

## Current Game State
[ Current State Information ]
```

### B.1.4  LEDUC POKER

```
# Leduc Poker Decision Prompt

You are playing **Leduc Poker** with 12 cards (0 0 1 1 2 2 3 3 4 4
    5 5).
You are **Player [Player ID]**.

---

## Game Rules
1. **Deal private cards:** each player is randomly dealt 1 private
   card from the deck (hidden from the opponent).
2. **Blinds:** each player posts a blind of 1 chip.
3. **First betting round:** Player 0 acts first. Actions include:
   - 'fold' (give up the hand)
   - 'call' (match the current bet)
   - 'raise' (increase the bet)
   The legal actions will be explicitly provided each turn.
   - In Round 1, there may be up to **2 raises**, each raise is
     fixed at **2 chips**.
4. **Reveal public card:** one public card is drawn randomly from
   the remaining deck.
```

```
5. **Second betting round:** same rules as Round 1, except the
   fixed raise amount is **4 chips**.
6. **Showdown:** if no one folds, players reveal hands:
   - If a player's private card pairs with the public card, that
     hand wins.
   - If neither player makes a pair, the higher private card wins.
   - Card ranking: 5 > 4 > 3 > 2 > 1 > 0.
   - A tie is possible.

---

## Current State
- Your private card: [ Private ]
- Current round: [ First Round / Second Round ]
- Public card: [ Public / None yet ]
- Round 1 betting sequence: [ Round1 ]
- Round 2 betting sequence: [ Round2 ]
- Legal actions: [ Legal Actions ]

---

## Action Instruction
Based on the current state and rules, choose **one legal action**
   as your decision.
```

### B.1.5    LIARS DICE

```
# Liars Dice Decision Prompt

You are playing a **two-player Liars Dice** game.

---

## Game Rules
1. Each player has **5 six-sided dice**, visible only to
   themselves.
2. Players take turns and may choose to either **make a bid** or
   **call "Liar"**.
3. **Making a bid:** you declare that among all dice, there are at
   least *x* dice showing face *y*.
   - Format: "x-y" (e.g., "3-4" means "at least three 4s").
4. Each new bid must be **higher** than the previous one (either a
   larger quantity, or the same quantity with a higher face
   value).
5. Dice showing **1** are **wild** and can count as any face value
   .
6. **Calling "Liar":**
   - If the opponent's last bid is valid (considering wild 1s),
     you lose.
   - If the opponent's last bid is invalid, you win.

---

## Current State
- Your dice: [ Holes ]
- Action history: [ Action ]
```

```
---

## Action Instruction
Choose your action based on the current state.
You must select exactly **one** of the following formats:
1. "Liar": challenge the opponent's last bid.
2. "x-y": make a new bid that is strictly higher than the previous
    one.
```

### B.1.6 TABOO WORDS

```
# Taboo Words Decision Prompt

You are **Player 1 (hachi)**. Your opponent is **kichii**.
You are now playing a **Taboo Words** game.

---

## Personality Traits
- Calm and highly strategic.
- Outwardly polite and composed, speaking clearly and logically.
- Internally, always calculating how to guide the opponent.
- Rarely shows emotions; even when the opponent is close to saying
    the taboo word, you remain composed and avoid revealing
  excitement.

---

## Game Rules
- The current conversation topic is **party arrangements**.
- The opponent's taboo word is **"eat"** .
- If you can successfully guide the opponent to say **"eat"**, you
    win.

---

## Speaking Rules
- Players take turns speaking.
- Each turn lasts **15-60 seconds** (about 10-400 characters).
- You may state your ideas or respond to the opponent.
- Exceeding the time limit, speaking too briefly, or staying
    silent all count as violations.

---

## Win/Loss Mechanism
- If multiple rounds are played, the player who accumulates more
    losses is the final loser.

---

## Conversation History
[ Dialogue so far ]
```

## B.2 ELO SCORE CALCULATION

For games with only win/loss outcomes, Elo ratings are updated directly. Suppose player $A$ has rating $R_A$, player $B$ has rating $R_B$, and their rating difference is $D = R_A - R_B$. The probability that player $A$ defeats player $B$ is

$$E = \frac{1}{1 + 10^{-D/400}}. \tag{16}$$

If the actual outcome is $S$ (win = 1, loss = 0, draw = 0.5), the rating update for player $A$ is

$$R_A := R_A + K(S - E), \tag{17}$$

where $K = 0.2$ is the rating adjustment parameter.

In contrast, games such as Texas Hold'em and Leduc Poker yield average payoffs (e.g., losing $0.5$ chips per hand), which cannot be directly mapped to win probabilities. In these cases, given the expected payoff $\mu_A$ and variance $\sigma_A^2$ of player $A$ against player $B$, the win probability with initial stack $L$ is approximated by

$$P\{\text{win with stack } L\} = 1 - \frac{1}{1 + \exp\left(\frac{2L\mu_A}{\sigma_A^2}\right)}. \tag{18}$$

Since only sampled estimates $\hat{\mu}_A$ and $\hat{\sigma}_A^2$ are available, small sample sizes can cause instability. To mitigate this, we adopt a smoothed formula:

$$P\{\text{win with stack } L\} = 1 - \frac{1}{1 + \exp\left(\frac{2L\mu_A}{\max(\sigma_A^2, 100)}\right)}, \tag{19}$$

which prevents extreme values and stabilizes estimation.

## B.3 INVALID POLICY ADJUSTMENT

LLMs frequently produce invalid moves. Even strong models such as o3 make illegal actions in roughly 10% of Texas Hold'em trials, with a common case being folding at the start of a street. Chess exhibits even higher error rates. To account for this, outcomes are discounted. The adjustment is proportional to task complexity: in games with richer action spaces or complex formats, errors are more tolerable, while in simpler games, errors are treated as near losses. For example, in Chess, an invalid move is penalized as $0.3$ of a loss. The factors used in this study are:

| Game | Error Adjustment |
|---|---|
| Leduc Poker | 1.0 |
| Taboo Words | 0.7 |
| Liars Dice | 1.0 |
| Texas Hold'em Poker | 0.7 |
| Goofspiel | 0.7 |
| Chess | 0.3 |

Table 3: Error adjustment factors for different games.

As models improve, invalid actions can eventually be treated as outright losses.

## C   TEXAS HOLD'EM WEIGHT DESIGN

In this study, the reward model is designed to measure the discrepancy between the ideal policy and the LLM-generated policy. A larger distance results in a smaller reward, while a smaller distance yields a larger reward. At the initial stage, in order to compare the action distribution of GPT-5-mini with that of the initial model, the Kullback–Leibler (KL) divergence can be adopted as the metric, since KL divergence naturally quantifies the difference between two distributions:

$$R(\omega_\theta, I, \omega^*(I)) = -\mathbb{D}_{KL}(\omega_\theta(I), \omega^*(I)) \tag{20}$$

Here, $\mathbb{D}_{KL}(\cdot\|\cdot)$ denotes the KL divergence between two strategies. However, it is important to note that different actions have different levels of strategic importance. For instance, in Texas Hold'em, the strategic significance of *Fold*, *Call/Check*, *Raise*, and *Allin* is not equivalent. Since KL divergence implicitly assumes equal importance for all actions, it can cause the LLM to overlook errors in crucial actions such as *Allin*. To address this issue, we introduce manually designed weights that emphasize more critical actions. Specifically:

- When all four actions (*Fold, Call/Check, Raise, Allin*) are legal: $w = [3, 3, 5, 10]$;
- When only three actions (*Call/Check, Raise, Allin*) are legal: $w = [3, 5, 7]$;
- When only two actions (*Fold, Call/Check*) are legal: $w = [7, 7]$;

This weighting design highlights the relative importance of different actions in strategy optimization, thereby enhancing the discriminative power of the reward function.

## D   LLM COACH

Our LLM Coach uses GPT o3, and the number of data given to the LLM Coach each time is 15. The prompt of the LLM Coach is as follows:

### D.1   LLM COACH PROMPT

```
You are a professional Texas Hold'em coach. Your role is to
    synthesize [local board information], [opponent hand-range
    assumptions], [prior players' <think> outputs], and [
    simulation-based statistics for future scenarios] to deliver
    clear, actionable strategy advice.

Your guiding principle is to prioritize gradual, incremental
    improvements--each action refined by no more than 0.05--to
    build steady progress over time, and to avoid overcorrection
    driven by biased data or small-sample noise.

[Task Background (Fixed Environment)]
- Game info:
    - Each player starts with 50 chips (25BB)
    - Small blind = 1, big blind = 2

- Turn order:
    - Player 0 = Big Blind (BB), Player 1 = Small Blind (SB)
    - Preflop: SB acts first
    - Flop/Turn/River: order reverses -- BB acts first, SB acts
        last

- Available actions:
    - '[Fold]' -- give up the hand
    - '[Call]' -- call the current bet (check is merged here)
    - '[Raise]' -- raise to 1.5x pot (fixed)
    - '[Allin]' -- go all-in with all chips

- You may choose only from the legal actions for each hand. The
    output must be a probability distribution strictly over those
    legal actions.
```

```
[Input Description (each hand has a uuid4_index identifier)]
- Basic info: position (IP/OOP), pot size, effective stacks (SPR),
    and action history
- Board: community cards (if any) and your hole cards
- Opponent clues: opponent's hole cards
- Simulation results: EV and confidence intervals for each legal
   action under fixed opponent hole cards

[Analysis Framework]
Your <think> must contain items 1)-3). Your <improved_action> must
    contain item 4) per uuid4_index.

1) Overall Diagnosis (multi-hand summary)
    - Identify stylistic biases: too passive/aggressive, over-
       calling, over-folding, unbalanced value-to-bluff ratios.
    - Compare action frequencies with the opponent's line and the
       board structure; highlight systematic mismatches (e.g.,
       under-c-betting on dry boards, overfolding on dynamic
       boards).

2) Grouped Per-Hand Key Points
    - Cluster hands into categories rather than analyzing them
       one by one. Suggested tag sets:
        1. Hand Strength (nuts, strong hand, showdown value,
           draw, air)
        2. Board Texture (dry/static vs. wet/dynamic)
        3. Opponent Range/Line (capped/uncapped, polarized/
           condensed, strong vs. wide line)
        4. Equity & Risk (SPR, IP/OOP, realization potential)
    - For each cluster, provide representative tags and concise
       key points.

3) Future Improvement Directions (per cluster)
    - Judge adjustments based on the aggregated EV and 95% CI
       patterns across the entire cluster, not on single-hand
       anomalies.
    - Remember that the simulation EVs are calculated against
       fixed opponent hole cards; broaden the perspective to
       account for the opponent's full plausible range (range-
       weighted correction, blockers, distributional fragility).
    - Prefer "robust ordering": if CIs overlap, lean toward
       actions with stronger equity realization, board coverage,
        or information gain.
    - In OOP spots, lean conservative.
    - In multi-way pots, adopt a more conservative strategy; in
       heads-up pots, play more aggressively when supported by
       EV/CI evidence.
    - If both the current strategy and the simulation outputs
       look severely distorted, override them and enforce the
       correct action. Such overrides must remain rare (<=10% of
        all cases).

4) Improvement Suggestions (per uuid4_index, output in <
    improved_action>)
    - Output must be a final probability distribution over only
       the legal actions, reported with two decimals and summing
        to 1.
    - Adjustment discipline:
```

```
            1. Per-action cap: no single action may change by more
               than 0.05. This small adjustment limit makes
               learning easier to follow and keeps your strategy
               steady.
            2. Stability: if the current strategy is already solid,
               prefer no change or only micro-adjustments (<=0.03).
            3. Dominated actions: if a strategy is strictly
               dominated, set its probability to 0 (no residual
               frequency).
            4. All-in discipline:
                  - Any probability change >0.03 in a single update
                     is invalid and will be rejected.
                  - Increasing the probability of allin requires
                     strong, reproducible evidence.
                  - The advantage must not depend on a tiny subset
                     of inflated combos.

[Output Format]
- First output <think> ... </think> containing:
    1. Overall diagnosis (multi-hand summary)
    2. Grouped per-hand key points
    3. Future improvement directions (per cluster)

- Then output <improved_action> ... </improved_action>, containing
   only:
    uuid4_index: [p1, p2, ...]      # probabilities for the legal
         actions, sum to 1; two decimals

- Do not add any extra text outside <think> and <improved_action>.

---
<think>
    write here:
    1) Overall diagnosis;
    2) Grouped key points;
    3) Future improvement directions
</think>

<improved_action>
    uuid4_index_1: [p1, p2, ...]
    uuid4_index_2: [p1, p2, ...]
    ...
</improved_action>
---
```

## D.2 EXAMPLE OF LLM COACH OUTPUT

```
<think>
1) Overall diagnosis
The current strategy set is broadly sound but shows three
   recurring leaks:
- Excess passive calling in spots where either (i) a medium-sized
   value / bluff raise or (ii) a polarised all-in is clearly
   higher-EV.
- Mild over-representation of dominated actions (e.g. negative-EV
   small raises, or keeping a residual 10 % all-in frequency when
    EV is deeply negative).
```

```
1188   - On flops/turns we under-protect against large bets with strong
1189       made hands (e.g. AA on Q-J-5 rainbow) - the nuts and near-nuts
1190        were not pressing enough edge.
1191
1192   2) Grouped per-hand key points
1193   A. Premium / near-nuts, low SPR (4a3b..., 677fcc... post-raise)
1194       ...
1195
1196   B. Strong draws / high card equity where fixed-bet all-in exploits
1197        villain's capped range (d4783..., 3208b..., 8d5b2a..., 51d74b
1198        ...)
1199       ...
1200
1201   C. Marginal pre-flop holdings facing limp / min action (4656f...,
1202      31a7c..., 0040c...)
1203       - Tags: showdown value, high realise potential, wide ranges.
1204       - Key: favour cheap calls; trim folds/raises that give up
1205          equity or create bloated pots.
1206
1207   D. Clearly negative spots (1e0dc..., 771931..., 176fab..., 24385b
1208        ...)
1209       ...
1210
1211   3) Future improvement directions
1212   - Range-weighted sanity check before trusting single-hand sims:
1213      only nudge frequencies <=0.1 unless EV gap is large & CI non-
1214      overlapping (> 2\sigma).
1215   - All-in discipline honoured: max upward +0.10; big downward cuts
1216      only when action is dominated.
1217   - Post-flop OOP: when EV of raise > call with non-overlapping CI,
1218      push raise up to 0.45 ceiling but keep call backbone >=0.25 to
1219       protect vs. range shifts.
1220   - Pre-flop vs. limp line: default keep at least 50 % call unless
1221      holding is premium or trash with fold option.
1222   - Continue monitoring dominated-action residue; target <5 %
1223      overall.

1224   </think>
1225   <improved_action>
1226   ...
1227   ...
1228   0040c565-adbb-44e0-8812-16089bbc775c: [0.75,0.20,0.05]
1229   ...
1230   ...
1231   </improved_action>
```

First, LLM Coach's analysis logic is very general, consistent with the playing style of Texas Hold'em, and it uses a lot of terminology. More importantly:

- 0040c565-adbb-44e0-8812-16089bbc775c: This action was classified as belonging to a *marginal hand* category, where the strategy should lean toward calling rather than excessive raising. The policy was updated from [0.6, 0.3, 0.1] to [0.75, 0.2, 0.05]. However, simulation results [-1.4, 5.0, -6.6] indicated that raising yields a higher payoff. This illustrates that when acting as a coach, the LLM does not fully rely on simulation outcomes but instead incorporates its own judgment.

# E SUPPLEMENTARY EXPERIMENTS

## E.1 TEXAS HOLD'EM SUPPLEMENTARY EXPERIMENTS

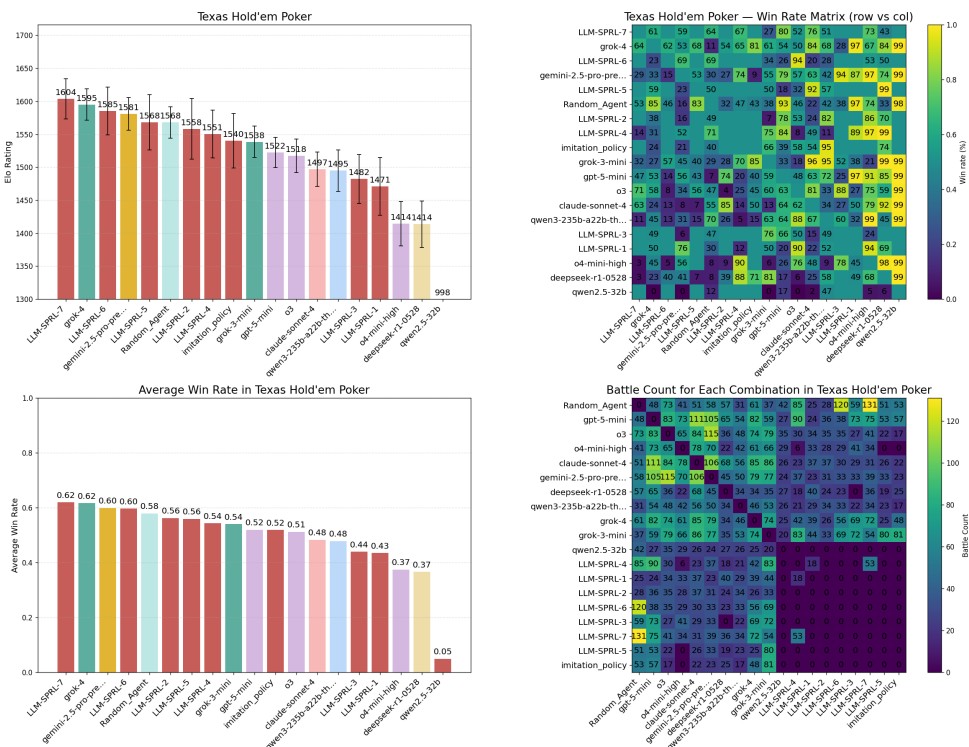

Figure 5: Texas Hold'em

## E.2 OTHER GAME SUPPLEMENTARY EXPERIMENTS

# F LLM USAGE

In preparing this paper, we used large language models (LLMs) as auxiliary tools for language refinement and programming assistance. Specifically, LLMs were employed to polish the presentation of text for clarity and conciseness, and to provide debugging support and code completion for standard experimental scripts. No part of the research ideas, theoretical results, or experimental designs was generated by LLMs. The authors take full responsibility for the content of this paper.

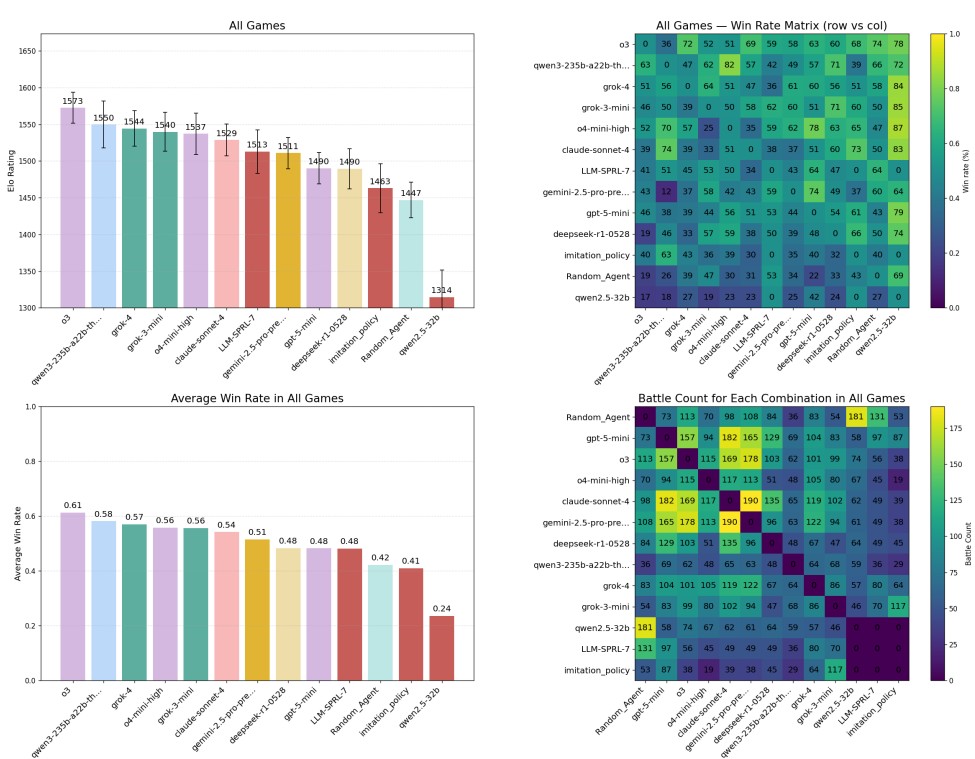

Figure 6: All Game

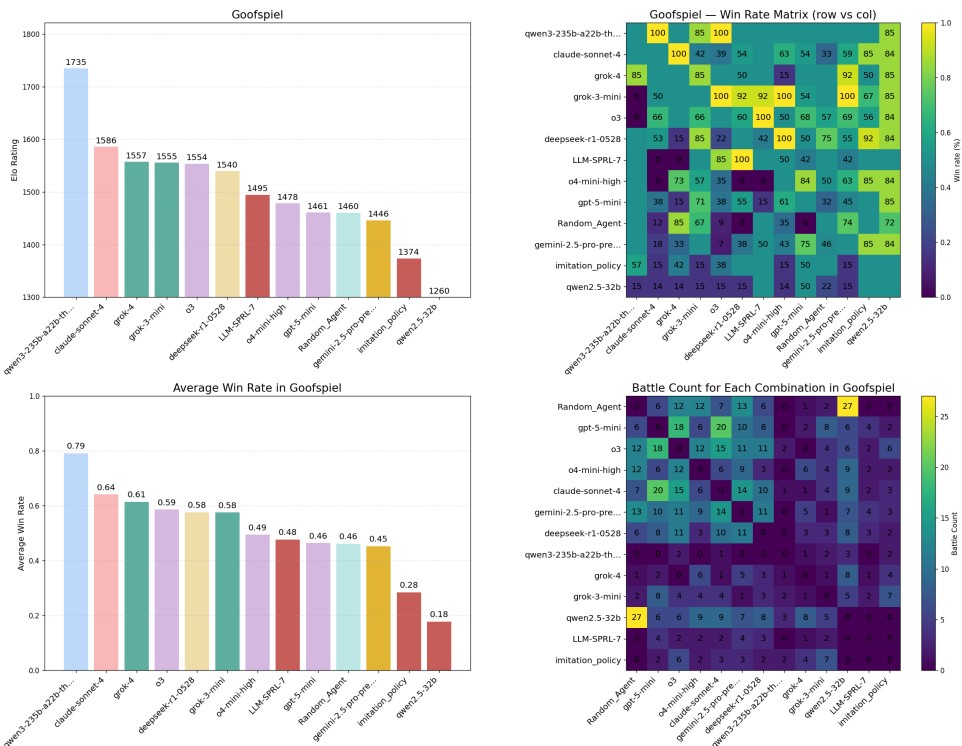

Figure 7: Goofspiel

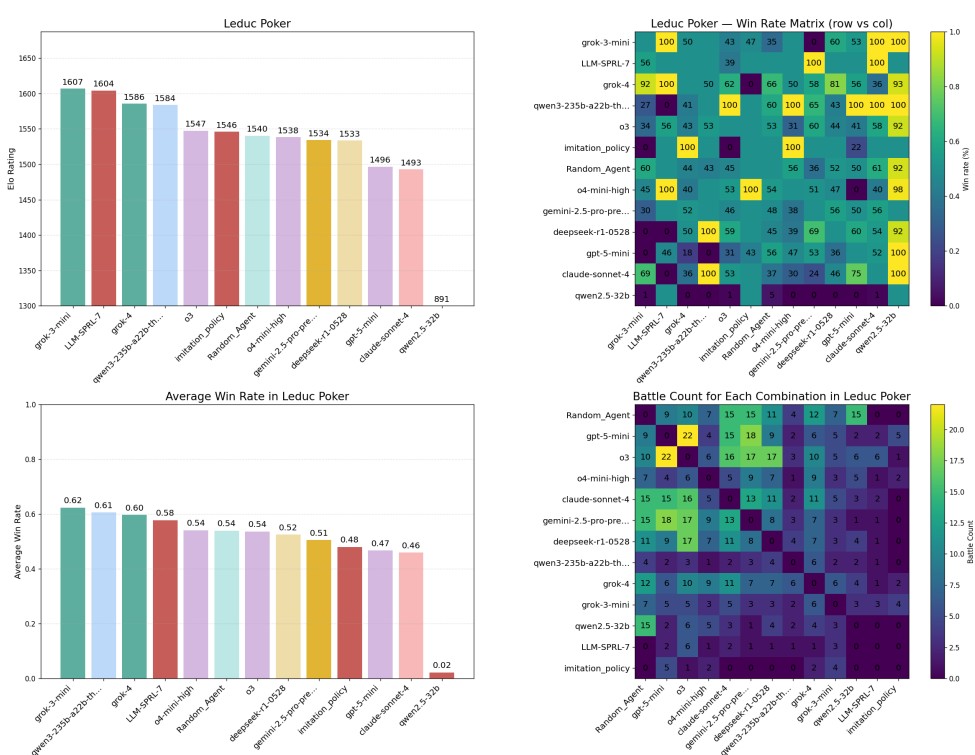

Figure 8: Leduc Poker

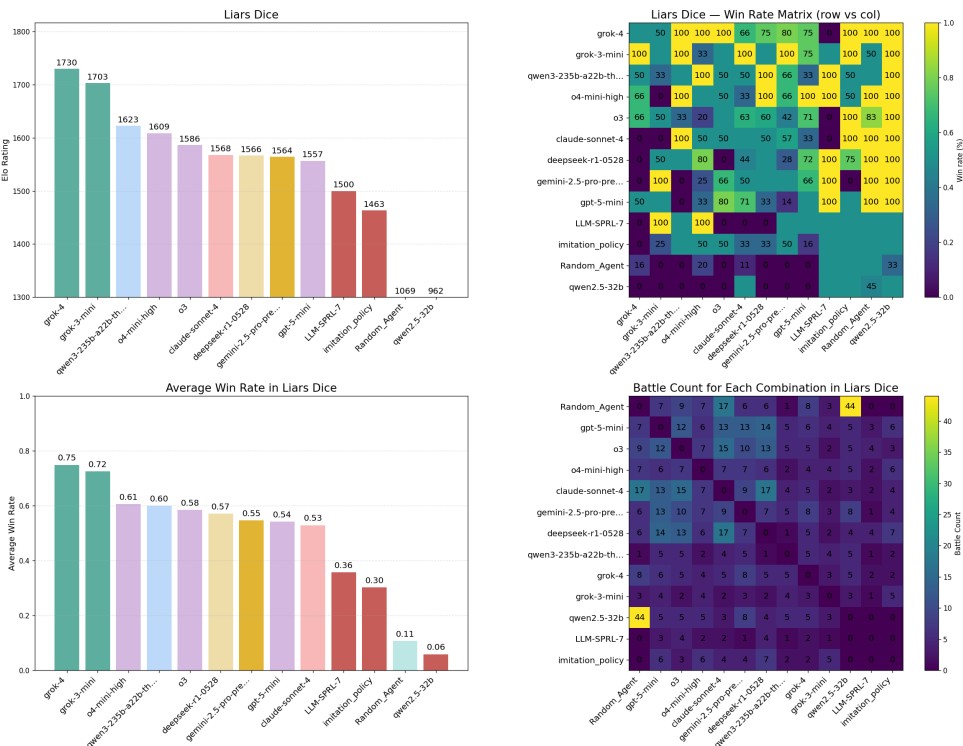

Figure 9: Liars Dice

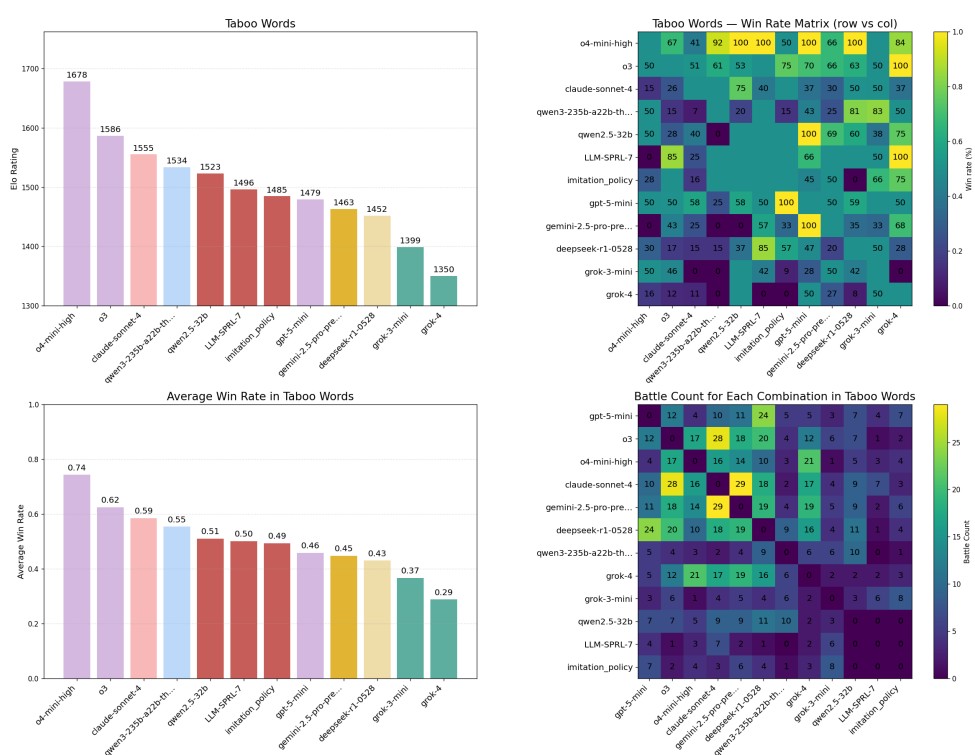

Figure 10: Taboo Words

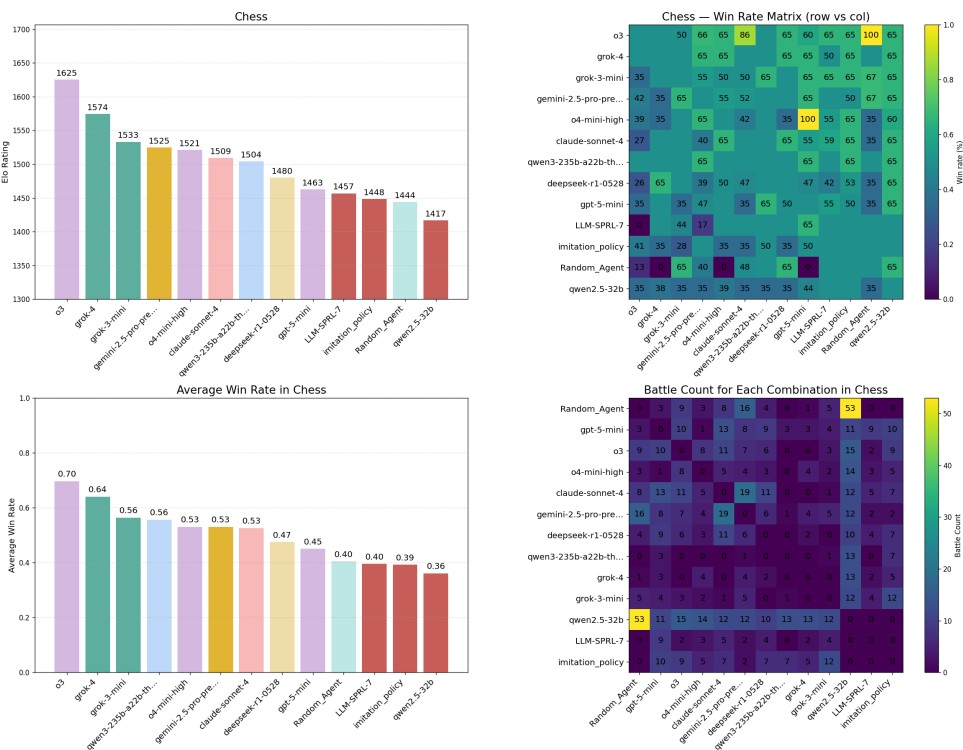

Figure 11: Chess

