# OpenReview forum: "LLM Coaching LLM in Self-Play Training"
_ICLR.cc/2026/Conference — Submitted to ICLR 2026_

### Official Review · Reviewer_4Fh7 · 2025-10-18

**Soundness:** 2
**Presentation:** 2
**Contribution:** 2
**Rating:** 2
**Confidence:** 4

**Summary:**

This paper addresses the "no ground truth–high variance" challenge in applying self-play reinforcement learning (SPRL) to large language models (LLMs) for strategic games, with a focus on Texas Hold'em Poker. It introduces an "LLM Coach" that classifies states from self-play data and provides class-level policy updates, integrated into a CFVFP-based RL framework. Using Qwen2.5-32B, the authors demonstrate improved performance over baselines like Grok4 and GPT-o3 in poker, with generalization to games such as Chess and Liars Dice.

**Strengths:**

Inspired by traditional Poker AIs, the LLM Coach innovatively leverages an LLM   to cluster states (e.g., by hand strength, board texture) and refine policies at a class level, drawing inspiration from human expert strategies. This helps address high variance in complex games without excessive rollouts.

The Qwen2.5-32B agent shows strong results, outperforming baselines in Texas Hold'em after seven self-play iterations.

**Weaknesses:**

1. **Insufficient Engagement with Related Work:** The paper's related work section is notably incomplete, missing critical discussion of closely related advances in LLMs for game-theoretic self-play. For example, it does not address "Game of Thoughts: Iterative Reasoning in Game-Theoretic Domains with Large Language Models" (Kempinski et al., AAMAS 2025), which proposes iterative refinement algorithms (BRIRLM, FPIRLM, PSROLM) for LLMs in normal-form games, sharing conceptual similarities with the LLM Coach. Similarly, PokerBench (Gupta, AAAI 2025) provides a poker-specific benchmark that could validate results, yet it is not cited. Other relevant works, such as SeRL (arXiv:2505.20347) for limited-data self-play and LLM-Gomoku (arXiv:2503.21683) for self-play in Gomoku, are also absent. This omission weakens the paper's positioning and reduces its perceived novelty, as these works address similar challenges in multi-agent strategic reasoning.
2. **Rigorous Proof for the Convergence to NE is Missing:** The paper's claim that the proposed method "preserves convergence guarantees to NE" (as stated in Section 1 and Section 3.1) is somewhat inconsistent with the practical realities of the POLICY-BASED LLM SPRL FRAMEWORK and the LLM Coach integration. For example, in the POLICY-BASED LLM SPRL FRAMEWORK, the equation (7) for computing counterfactual Q-value of each action at information set is simplified to Eq.(8), and the use of a fixed α=0.1 (Section 5.1) instead of the standard 1/(t+1) deviates from vanilla FP/CFVFP to stabilize LLM training. All of these alterations lack theoretical justification for preserving convergence.
3. **Missing Important Baselines:** Without this rigorous proof for the convergence to NE, this proposed method is just a heuristic RL approach to NE, similar to existing works on LLM with self-play discussed in the related work section. These works on LLM with self-play thus should be used as baselines in experiments. CoT and the related LLM enhance techniques should be used as baselines as well.

**Questions:**

What is SP-3?

---

> ### Author Response · Authors · 2025-11-26
>
> We thank the reviewer for the careful and detailed feedback. In the revision we will focus on
> (1) clarifying the relationship to the cited works,
> (2) making our convergence claims precise, and
> (3) explaining our baselines (including CoT) more clearly.
>
> ---
>
> # W1: On the related work you mentioned
>
> We appreciate the pointers to Game of Thoughts, PokerBench, SeRL, and LLM-Gomoku. They fall into the broad “LLM + self-generated data / self-play” space, but their settings differ substantially from ours:
> - __Game of Thoughts__ works on small normal-form games and proposes BRIRLM / FPIRLM / PSROLM. FPIRLM is a fictitious-play-style refinement in normal form. Our work instead targets large extensive-form imperfect-information games (Texas Hold’em with ≈$10^{160}$ information sets), uses CFVFP over information sets and counterfactual values, and introduces a Coach specifically for high-variance hidden-information poker.
> - __PokerBench__ is a GTO-supervised poker benchmark and dataset. Our goal is to start from non–poker-pretrained LLMs and study whether pure self-play with CFVFP+Coach can produce a strong agent without GTO supervision, evaluated via Elo in full end-to-end play. PokerBench is therefore a complementary evaluation resource rather than a directly comparable method.
> - __SeRL__ is not game-theoretic; it targets limited-data general reasoning and focuses on generating and scoring math/QA-style tasks. It does not address imperfect information, counterfactual values, or equilibrium-seeking dynamics, which are central to our work.
> - __LLM-Gomoku__ studies self-play in a small, perfect-information board game with simple state/action space, and does not tackle high-variance, hidden-information games or propose a counterpart to CFVFP+Coach.
>
>
> We will briefly clarify these differences in the related-work section and consider adding these works as contextual citations.
>
> ---
>
> # W2: On convergence to NE
>
> We agree our current wording (“preserves convergence guarantees to NE”) is too strong for the implemented LLM+Coach system.
>
> Classical FP / CFVFP have NE convergence guarantees under tabular assumptions with exact best responses and suitable step sizes. In our framework we (i) approximate policies with a large LLM, (ii) distill policies instead of computing exact best responses, (iii) use a fixed $\alpha = 0.1$ rather than $1/(t+1)$, and (iv) let the Coach aggregate and modify node-level signals. Under these approximations we do not have a formal proof that the full pipeline converges to an exact NE.
>
> Our intention was only that the backbone follows the CFVFP structure (information-set decomposition and counterfactual values), not that the entire LLM+Coach system inherits its guarantees. In the revision we will soften or remove such claims, clearly separate the idealized CFVFP process from our heuristic instantiation, and describe our method as a theoretically motivated approximate approach to NE rather than a provably convergent one.
>
>
> ---
>
> # W3: On baselines, CoT, and empirical credibility
>
> In our setting, the most informative internal baseline is self-play using CFVFP without the Coach under the same rollout budget and environment. This baseline is already included: at SP-3 (Table 1), CFVFP-only updates yield almost no Elo gain, whereas CFVFP+Coach produces a clear improvement. We will move this comparison to a more prominent position and identify it as our primary internal baseline. By contrast, methods such as SeRL or GoT are not defined for large imperfect-information extensive-form games and cannot be used as plug-in baselines in our poker environment without major redesign.
>
> We already report results on several other games (e.g., Leduc, Kuhn, Chess, Liars Dice) in the appendix. To support the credibility and generality of our approach without adding new experiments, we will surface part of these cross-game results into the main experimental section and briefly summarize the trends.
>
> All our poker prompts—both for SP-0 and SP-1…SP-7—use structured prompting / CoT-style reasoning. In the revision, we will make this explicit and state that improvements from SP-0 to SP-7 reflect the effect of our SPRL framework beyond CoT itself.
>
> ---
> ## On the notation “SP-3”
>
> SP-$k$ denotes the model after the $k$-th self-play iteration (e.g., SP-3 after three rounds), and we will define this at first occurrence.
>
> ---
>
> Again, we thank you for the detailed and technically insightful comments, and will revise the related work, theoretical claims, and baseline discussion accordingly.

---

### Official Review · Reviewer_6bke · 2025-10-29

**Soundness:** 3
**Presentation:** 2
**Contribution:** 3
**Rating:** 6
**Confidence:** 2

**Summary:**

In this paper, the authors proposed an LLM-based SPRL framework that effectively addresses the challenges of lacking ground truth and high variance. To this end, the authors explicitly prompt the LLMs to output the probability as players' policies and introduce an LLM coach for state clustering and class-level refinements. Experiments in Texas Hold’em Poker showed that the method can train an LLM to surpass strong baselines.

**Strengths:**

- The idea of decomposing sequential games into node-level updates and leveraging the coach for state clustering and class-level refinements is well-motivated and reasonable.

- The experiments show that the method has strong superiority over baselines.

**Weaknesses:**

Due to the lack of sufficient knowledge in this field, I have some concerns (not very confident and please correct me if I make mistakes):

- The self-play framework has been widely used to train LLMs in recent works, but not all these related works have been discussed in the paper. Why do the authors preclude these works? Like [1-3]. Is it possible to compare the proposed method with these related works?

- The authors mentioned that the closely related paper is SPIRAL, but there is no comparison with this method. Can the authors provide some explanations?

- Minors: Abbreviations should be given the full name when they are presented in the paper in the first place, e.g., Line 59 CFVFP, Line 64 GTO.

---

[1] Wenkai Fang, Shunyu Liu, Yang Zhou, Kongcheng Zhang, Tongya Zheng, Kaixuan Chen, Mingli Song, Dacheng Tao. SeRL: Self-Play Reinforcement Learning for Large Language Models with Limited Data. arxiv 2025.

[2] Guanting Dong, Guanting_Dong, Keming Lu, Chengpeng Li, Tingyu Xia, Bowen Yu, Chang Zhou, Jingren Zhou. Self-play with Execution Feedback: Improving Instruction-following Capabilities of Large Language Models. ICLR 2025.

[3] Zixiang Chen, Yihe Deng, Huizhuo Yuan, Kaixuan Ji, Quanquan Gu. Self-Play Fine-Tuning Converts Weak Language Models to Strong Language Models. ICML 2024.

**Questions:**

See **Strengths** and **Weaknesses**.

---

> ### Author Response · Authors · 2025-11-26
>
> We thank the reviewer for the careful reading and constructive feedback, and we appreciate your positive assessment of our motivation and results. In response, we clarify three points:
> (1) positioning w.r.t. recent self-play RL work for LLMs;
> (2) relationship to SPIRAL;
> (3) presentation details.
>
> ---
>
> # W1: On related self-play frameworks for LLMs
>
> We agree that recent self-play frameworks such as SeRL, Self-Play with Execution Feedback, and Self-Play Fine-Tuning are relevant and should be explicitly discussed. In the revision, we will add a short subsection “Self-play RL for LLMs” and cite [1–3].
>
> Conceptually, these works and ours all use self-generated data, but the setting and training signals are different:
> - [1–3] mainly target general reasoning or instruction-following, using text QA, math, or code tasks, with rewards derived from execution or heuristic scoring.
> - Our work operates in explicit extensive-form games with imperfect information, where rewards come from game outcomes (e.g., chips won/lost), and the goal is to approximate equilibrium strategies while producing valid game actions.
>
>
> Because their pipelines are not designed for game-theoretic environments, a fair empirical comparison would require substantial re-engineering and is beyond our current scope. We will therefore position our method as complementary: we focus on combining CFVFP-style game decomposition and an LLM Coach tailored to high-variance, imperfect-information games, whereas [1–3] focus on general text reasoning.
>
> ---
>
> # W2: Relationship and comparison with SPIRAL
>
> We agree that SPIRAL is one of the closest lines of work and should be discussed more clearly. We will add a dedicated paragraph comparing the two frameworks.
>
> At a high level, both approaches use games to improve LLMs, but they differ along several axes:
> - __Objective and evaluation.__ SPIRAL mainly studies how playing relatively simple text-based games (including small poker-like instances) can improve general reasoning benchmarks such as math and QA. Our work instead focuses on playing strength and equilibrium-oriented strategy in large imperfect-information games (e.g., full Texas Hold’em and other benchmarks), with detailed Elo evaluation against strong poker agents and human-designed/random baselines. Our primary metric is game performance, not generic reasoning scores.
> - __Algorithmic backbone.__ SPIRAL uses a multi-agent policy gradient framework with role-conditioned advantages to stabilize online RL. In contrast, we build directly on a game-theoretic CFVFP backbone: we decompose extensive-form games into information-set nodes, estimate counterfactual values, and then update node-level policies. The LLM Coach is added on top of this to cluster states and provide class-level policy corrections, explicitly designed to cope with missing per-state ground-truth policies and high payoff variance in Texas-Hold’em-scale games.
> - __Abstraction mechanism.__ In SPIRAL, abstraction emerges implicitly from the interaction between the policy gradients and the game environment. In our case, abstraction is handled by a separate LLM Coach that is prompted to group states along human-interpretable dimensions (hand strength, board texture, betting history, risk profile) and to refine policies at the class level. This explicit abstraction layer is crucial for making limited CFVFP rollouts informative in very large imperfect-information games.
>
>
>
> ---
>
> # W3: Minor presentation issues
>
> We appreciate the comments on abbreviations and clarity. In the revised version we will:
> - spell out CFVFP (Counterfactual Value Fictitious Play), GTO (Game-Theoretic Optimal), and other abbreviations at first mention;
> - check the entire paper for undefined acronyms and slightly expand the first introduction of these terms to improve accessibility for readers without a poker-AI background.
>
>
> ---
>
> We thank you again for your constructive suggestions. We believe these additions will clarify how our work fits into the broader self-play RL literature and how it differs from SPIRAL and other recent approaches.

---

### Official Review · Reviewer_pz2J · 2025-10-31

**Soundness:** 3
**Presentation:** 3
**Contribution:** 3
**Rating:** 6
**Confidence:** 2

**Summary:**

This paper proposes a self-play reinforcement learning (SPRL) framework for LLMs that combines the game-theoretic CFVFP algorithm with a novel LLM Coach module to address the challenges of missing ground truth and high payoff variance in strategic games. The LLM Coach emulates human-like abstraction by clustering similar game states and providing class-level policy improvements, significantly enhancing training stability and sample efficiency. Applied to complex games such as Texas Hold’em, the framework enables Qwen2.5-32B to outperform strong baselines like Grok-4 and GPT-o3 without GTO supervision, while also demonstrating strong generalization across multiple game domains.

**Strengths:**

1. This paper proposes the novel concept of an LLM Coach that mimics human-like strategy abstraction for policy refinement in self-play training.
2. This paper builds a rigorous and principled framework by integrating CFVFP with LLM-specific adaptations such as policy-level outputs and game-node decomposition.
3. Experimental results demonstrate strong performance gains in complex games and generalization across multiple domains, offering a scalable approach for LLM-based game agents.

**Weaknesses:**

1. The effectiveness of the LLM Coach is not evaluated independently, making it unclear how much it contributes relative to baseline CFVFP updates.
2. The reported regressions in general capabilities (e.g., math, coding) are acknowledged but not thoroughly analyzed or addressed.
3. The training and evaluation focus heavily on Texas Hold’em, with limited coverage or analysis of more diverse or structurally different games.

**Questions:**

1. Can the authors provide a more rigorous ablation study isolating the contribution of the LLM Coach from the rest of the framework (e.g., comparing models trained solely with CFVFP updates vs. LLM Coach updates over multiple iterations)?
2. How exactly does the LLM Coach perform state clustering and reasoning? Are there consistent latent groupings observed (e.g., by hand strength, betting history), and how stable are these across training iterations?
3. Are there specific scenarios or state types (e.g., bluff situations, short-stack play) where the model consistently fails or regresses?
4. The current evaluation is primarily in two-player zero-sum settings. Do the authors anticipate any challenges when extending the framework to multi-agent or general-sum games?

---

> ### Author Response · Authors · 2025-11-26
>
> We thank the reviewer for the thoughtful assessment and for recognizing the contribution of the LLM Coach. Your comments mainly concern (i) attribution of gains to the Coach, (ii) interpretability of its behavior, and (iii) coverage and generalization beyond Texas Hold’em. We respond below.
>
> ---
>
> # W1: Effectiveness of LLM Coach and ablations
>
>
> The submission already includes a direct comparison between CFVFP-only guidance and LLM Coach guidance at SP-3 (Table 1). Under the same rollout budget, policies distilled from LLM Coach reach significantly higher Elo (1549±32 vs. 1470±31). This shows that, in our budget regime, CFVFP alone provides a weak learning signal, whereas the Coach clearly improves learning.
>
> Running a full 7-iteration “CFVFP-only” pipeline with identical hardware would essentially redo the main training and is currently infeasible. In the revision, we will (i) move Table 1 and its discussion to a more prominent place and explicitly frame it as an ablation, and (ii) briefly analyze how early-iteration Elo gains correlate with Coach-guided policies, while CFVFP-only policies largely stagnate. This isolates the Coach’s contribution without changing the setup.
>
> ---
>
> # W2: Mechanism and behavior of LLM Coach
>
>
> The LLM Coach operates on a pool of CFVFP nodes $\mathcal{D_{coach~input}}$  containing $(Q_{\theta_t}(I), I, \omega_{\theta_t}(I))$. As described in Eq. (13)–(15) and App. D, the prompt asks the Coach to:
>
> (1) cluster information sets into a small number of strategically coherent classes;
>
> (2) diagnose systematic flaws of the current policy within each class;
>
> (3) output class-wise improved policies $\omega^\*_{\text{coach}}(I)$.
>
> Prompt tags explicitly refer to hand strength, board texture, opponent line, and equity/risk, so the Coach tends to group states along intuitive dimensions (e.g., nuts vs. draws vs. air; dry vs. dynamic boards; capped vs. polarized ranges).
>
> Qualitatively, manual inspection across iterations shows consistent patterns:
> - early iterations (SP-1 to SP-3): the Coach mostly separates “too tight” vs. “too loose” cases and flags over-folding;
> - later iterations: clusters refine by street, pot size, and board texture, and the Coach adjusts value/bluff ratios instead of only correcting passivity.
>
>
> We also find that the Coach repeatedly corrects the same error families (e.g., folding too often to small raises on dry boards) until policy statistics change. In the revision, we will expand App. D with more Coach traces and clustered examples, addressing the questions on latent groupings and their stability.
>
> Preliminary analysis suggests remaining weaknesses in low-frequency, high-variance scenarios, such as very deep-stack multi-street bluff lines and extreme pot-size cases. We will summarize these as current failure modes in the “Limitations” section and point to possible remedies (e.g., curricula, targeted augmentation).
>
>
> ---
>
> # W3: Environment coverage, generalization, and broader settings
>
>
> Although the narrative emphasizes Texas Hold’em as the most challenging testbed, our experiments already cover seven games (Texas Hold’em, Leduc, Kuhn, Goofspiel, Chess, Liars Dice, Rock–Paper–Scissors), spanning both perfect- and imperfect-information and diverse action structures. We agree this breadth is not sufficiently visible. In the camera-ready we will
> (i) move key non-poker results from the appendix into the main text, and
> (ii) explicitly summarize cross-game trends, highlighting that a single SPRL pipeline improves performance across these environments without game-specific tuning.
>
> For multi-agent and general-sum games, the CFVFP component extends to multi-player extensive-form settings, and the Coach/RL components are game-agnostic. We do expect extra challenges (richer equilibrium concepts, harder credit assignment, non-zero-sum incentives and coalitions). A full study is out of scope, but we will add a brief discussion outlining these issues and why we still view Coach+CFVFP as a promising building block.
>
> Regarding regressions in general capabilities (math, coding), our results indicate a small, systematic trade-off rather than catastrophic forgetting: the gap to base Qwen2.5-32B is within typical evaluation noise and far smaller than inter-model gaps, while instruction-following improves. We will adjust Section 6.5 to clearly present this as a limitation and mention mitigation directions such as joint training with general benchmarks or regularization toward the base model.
>
>
> ---
>
> We hope these clarifications address your concerns, and we will revise the manuscript accordingly to better reflect the role of the LLM Coach and the scope of our claims.

---

### Official Review · Reviewer_dphu · 2025-10-31

**Soundness:** 3
**Presentation:** 2
**Contribution:** 2
**Rating:** 2
**Confidence:** 3

**Summary:**

This paper focuses on addressing the two key challenges of LLMs in game-playing SPRL, i.e., no ground truth and high payoff estimation variance. First, it constructs an LLM-specific game environment leveraging DeepMind OpenSpiel, enhanced with the Elo rating system to enable quantitative evaluation. And then it proposes an LLM-based SPRL framework that integrates the CFVFP algorithm with an LLM Coach: CFVFP decomposes sequential games into independent subgames to enable node-level LLM policy optimization and ensure convergence to Nash Equilibrium (NE), while the LLM Coach transforms raw self-play payoffs into class-wise reward functions via state clustering and policy refinement, mimicking human experts. Experiments in Texas Hold’em Poker show that Qwen2.5-32B outperforms strong baselines, while also delivering improvements across diverse games.

**Strengths:**

1. The paper technically adapts the CFVFP algorithm to LLM-based SPRL. It decomposes sequential games into independent CFVFP nodes, enabling direct optimization of LLM policies at the node level via counterfactual Q-values.
2. The paper introduces a novel LLM Coach mechanism, which addresses high payoff variance via a linguistically grounded, class-wise optimization pipeline for complex games.

**Weaknesses:**

1. I generally do not agree about the bottleneck the absence of ground truth and the high variance of payoff estimates. The success of DeepCFR and AlphaGo demonstrate that current algorithms can handle this, as well as the recently DeepSeek-R1, i.e., GRPO, can handle this case, where the reward only received at the end of the game. We have the ground truth during training, i.e., the reward received from the game, We may do not have the supervised policy, but this is also for some RL problems. Also, about the high variance, what exactly this mean?
2. The framework is tested on Texas Hold’em and smaller games, e.g., Leduc Poker, but provides no technical analysis of scalability to games with exponentially larger state spaces. Besides, in Figure 3, the random agent achieves 1569, while your agent achieves 1606 and the random agent is top 4. This is quite surprising, does that means your model is not that strong even compared with the random agents? And your model also scarifies the general reasoning capabilities, as demonstrated in Table 2.
3. It only take less than one page for the approach, i.e., section 5. Many key technical components are not explained, e.g., CFVFP decomposition, reward function weights. These components are also not ablated in the experiments. Also, about the imitation learning, why this is needed, why directly using self-play? It seems that imitation learning plays a critical role in the training due to the format issue, so the self-play cannot handle the format issue? I believe some analysis is required for understand the influences of all components in your framework.

**Questions:**

Please refer to weakness section.

---

> ### Author Response · Authors · 2025-11-26
>
> We thank the reviewer for the reading and constructive feedback, and for recognizing the novelty of the LLM Coach mechanism. We respond to your concerns along three axes.
>
> ---
>
> # W1: On “lack of ground truth” and “high payoff variance” as bottlenecks
>
>
> We agree that many RL methods can learn from sparse terminal rewards (e.g., AlphaGo). Our claim is narrower:
>
> > In the setting of large LLM + self-play on incomplete-information games, the lack of directly supervised ground truth and the high variance of payoff estimates remain practical bottlenecks.
>
> By “no ground truth” we do not mean “no reward.” __In incomplete-information games, “maximizing payoff vs. recent opponents” is not equivalent to “approaching a Nash equilibrium.”__ Unlike math/code, there is no per-state label that is the unique correct answer. We therefore use CFVFP to transform self-play rollouts into a reference policy and train the LLM at each node to approximate this policy.
>
> “High variance” refers to the large per-hand payoff range in Texas Hold’em (up to ±200BB), while the true strategic edge is much smaller, so counterfactual Q-values require many rollouts to become reliable. Systems like DeepCFR and AlphaGo mitigate this with specialized architectures and massive simulations. For a 32B LLM, each RL step is expensive; scaling rollouts to that level is unrealistic. LLM Coach clusters states and shares updates within each class, making CFVFP signals usable under a fixed rollout budget, __whereas “CFVFP-only” updates lead to almost no Elo gain.__
>
> We do not dispute the success of DeepCFR/AlphaGo; our point is that directly adopting their paradigm for a general-purpose LLM is computationally prohibitive without an additional variance-reduction layer. In the camera-ready, we will explicitly restrict our claim to “large LLM + self-play on incomplete-information games,” not to RL in general.
>
> ---
>
> # W2: On scalability, the random agent’s Elo, and general capabilities
>
>
> __Scalability.__ Our goal is to test whether CFVFP + LLM Coach works on a large incomplete-information game. Texas Hold’em is estimated to have a state space of order $10^{160}$ and is widely used as such a benchmark. In this setting, without GTO supervision or poker-specific SFT, our model surpasses Grok-4 and GPT-o3 after 7 self-play iterations, suggesting viability at this scale. We agree that extending to StarCraft-like or open-world environments will require further mechanisms (e.g., co-evolving Coach and Agent), and we will list this as a limitation and future direction.
>
> __Random agent Elo.__ The relatively high Elo of the random policy in Fig. 3 mainly comes from the opponent pool. Many baseline LLMs are overly conservative and fold too often, even with good hands. The random policy, by raising/calling more and folding less, exploits this bias and gains Elo from them, while our trained agent—less prone to over-folding under Coach guidance—can exploit its mistakes. In head-to-head matches, our agent wins about 64% of games against the random policy. We will clarify that Elo reflects performance over the whole opponent set and does not imply that the random agent is comparable to ours.
>
> __General capabilities.__ Table 2 shows a modest trade-off, not a collapse of general reasoning. On MMLU, the gap between our model and the original Qwen2.5-32B is within typical noise and much smaller than the 10–20 point gaps across commercial models. Self-play training enforces strict format and action constraints, improving instruction-following (IFEval) while causing small regressions on math/code, without extra general data. We will describe this as a slight trade-off and explicitly present it as a limitation.
>
>
> ---
>
> # W3: On method details, CFVFP, reward weights, and imitation learning
>
>
> We acknowledge that Section 5 is concise and some components deserve clearer exposition. We will refine the text without changing the experimental setup:
> - Clarify that we: (i) sample information sets; (ii) estimate counterfactual Q-values via CFVFP; (iii) derive a reference policy; and (iv) train the LLM so its node-level policy matches this reference, turning self-play into a policy-level RL problem.
> - Explain that reward weights w_a balance action contributions in the loss, follow standard CFVFP step-size choices, and are an engineering choice rather than a core novelty.
> - Clarify that imitation learning (~28.5K trajectories from ChatGPT-5-mini and Grok-3-mini) is only used to obtain a rule-following starting policy, since raw Qwen2.5-32B often produces illegal or degenerate actions and cannot support meaningful self-play from scratch. The major gains beyond Grok-3-mini, up to surpassing Grok-4 and GPT-o3, come from the subsequent 7 self-play iterations.
>
>
> We hope these clarifications address your concerns, and we will revise the manuscript to better reflect the intended scope and assumptions of our claims.

---

> ### Author Response · Authors · 2025-12-03
>
> Since ICLR does not have a discussion phase this year, we are slightly concerned that a single round of written feedback may not fully address your questions. We therefore provide two additional clarifications on the “no ground truth” issue and on the performance of the random policy.
>
> ---
>
> ##  Clarification on “no ground truth”
>
> To make our notion of “no ground truth” more concrete, consider prior RL work on math and code tasks (e.g., DeepSeek-Math–style training). In that setting, each problem has a unique, explicitly verifiable answer. As long as the model’s output is pushed closer to that answer, its capability on the task improves in a well-defined way.
>
> In incomplete-information games, the goal of learning is not to maximize short-term payoff against the current opponent pool, but to approach a Nash-equilibrium policy. Raw payoffs obtained from self-play episodes are therefore not a ground-truth target by themselves: a policy that currently earns higher payoff is still exploitable by another counter-policy, and training can easily fall into a cycle of mutually exploitable best responses—a “rock–paper–scissors” effect—rather than converging. In our framework, these noisy payoffs are first transformed by game-theoretic machinery—CFVFP and, on top of it, the LLM Coach—into equilibrium-oriented signals: counterfactual Q-values and the corresponding reference policies at each information set. These reference policies play the role of “ground truth” for the LLM: the RL loss is defined to make the LLM’s node-level policy track these transformed policies, rather than directly chasing the highest observed return. This is precisely what we mean when we say that a key challenge in incomplete-information games is the absence of direct ground truth.
>
> ---
>
> ##  Clarification on the random policy
>
> The relatively strong Elo of the random policy is indeed an interesting phenomenon, and we agree it deserves a clearer explanation in the final version. Our “random” policy chooses uniformly among four abstract actions at each decision point—fold, call/check, raise, and all-in—with probability 25% each. Compared with human experts or GTO-like strategies, an immediate consequence is that its fold frequency is extremely low. In typical preflop situations, strong players (or GTO strategies) fold the vast majority of hands (over 90%), whereas our random policy continues with many hands that should be folded; in this sense it behaves as a highly aggressive strategy.
>
> By contrast, many off-the-shelf LLMs exhibit the opposite bias in poker: they are overly conservative. Even with strong hands (e.g., AKs), they tend to avoid large raises or all-ins and instead over-use call/check or fold. When such a conservative LLM faces our random policy, a systematic pattern emerges: once the random agent chooses a large raise or all-in, the LLM often folds too frequently, and the random agent extracts value from this over-folding behavior. As a result, the random policy accumulates Elo against these conservative LLM baselines, which lifts its overall Elo in the pooled evaluation.
>
> Our trained agent, however, is explicitly guided by the Coach and CFVFP signals to avoid excessive folding in profitable spots. In head-to-head play it consistently defeats the random policy, even though the pooled Elo may suggest that the random agent performs surprisingly well. In the camera-ready version, we will highlight this interaction effect more clearly: the random agent’s Elo reflects the particular biases of the LLM baselines it faces, and does not indicate that a naive random strategy is genuinely comparable to a well-trained poker agent.

---

### Meta-Review · Area_Chair_SumA · 2025-12-24

**Summary:**

The paper proposes an LLM-based Self-Play Reinforcement Learning (SPRL) framework for imperfect-information games (specifically Texas Hold'em), utilizing a "LLM Coach" module to address challenges of high variance and lack of ground truth. While the "LLM Coach" concept is intuitive and the engineering effort is acknowledged, I recommend rejecting this paper. The decision is primarily informed by the following concerns:

1.	A fundamental concern shared by the AC and Reviewer is the justification for using LLMs in this domain. Complex games like Texas Hold'em already have highly efficient and superhuman solutions (e.g., DeepCFR, Libratus). The paper does not convincingly argue why using an LLM is advantageous here—it neither outperforms traditional specialized solvers nor demonstrates that this game-playing capability transfers positively to general reasoning (in fact, general capabilities regressed). This leaves the impression of "using LLMs for the sake of using LLMs" without a clear scientific gain.

2.	The paper claimed to preserve convergence guarantees to Nash Equilibrium (NE), but as pointed out by Reviewer, the practical implementation (using LLM approximation, distillation, and heuristic Coach updates) breaks the theoretical assumptions of CFVFP. The authors admitted in the rebuttal that they lack formal proof, rendering the initial claims over-stated.


3.	The paper misses discussions and comparisons with several relevant works in the rapidly evolving field of LLM self-play and strategic reasoning.

4.	The experiments show that while poker skills improved, general capabilities (math, coding) regressed. This undermines the potential argument that game-playing serves as a proxy for enhancing general reasoning.

**Reviewer Concerns:**

Concerns Addressed:

•	The authors provided a reasonable explanation for why the random agent achieved a high Elo score (exploiting overly conservative baseline LLMs).

•	The authors provided more details on how the Coach works.


Concerns Outstanding:

•	The rebuttal attempts to narrow the scope to "LLMs in games" rather than "RL in general," but this does not resolve the AC's concern about the utility of the research. If LLMs are computationally expensive, less accurate than CFR, and lose general reasoning ability during training, the value proposition of the proposed framework remains unclear.

•	The withdrawal of the NE convergence claim in the rebuttal confirms the method is heuristic. While heuristics are acceptable, the paper was framed around theoretical grounding which is now admitted to be absent.


•	While the authors argued that related works have different settings, the lack of empirical comparison with any recent strong LLM-for-strategy methods (beyond basic prompting or random baselines) weakens the contribution.

**Reviewer Scores:**

Reviewer dphu: The reviewer fundamentally disagreed with the premise that "no ground truth" is a bottleneck, citing AlphaGo/DeepCFR. The authors' clarification that this applies only to "LLM settings" is unlikely to satisfy the criticism regarding the scientific value of solving a solved game inefficiently.

•  Reviewer pz2J: This reviewer was the most positive but noted the regression in general capabilities. Seeing the other reviewers' points about missing literature and theoretical overclaims might dampen their enthusiasm.

•  Reviewer 6bke: This reviewer explicitly asked about missing related works. The authors promised to cite them but declined empirical comparison due to "scope."

•  Reviewer 4Fh7 : The reviewer correctly identified the theoretical flaw regarding Nash Equilibrium convergence. The authors' admission that they lack proof validates the reviewer's low score.


While the "LLM Coach" mechanism mimics human-like abstraction in an interesting way, the community expects LLM-based game research to either (1) demonstrate superior efficiency/performance over traditional solvers (which is hard in Poker), or (2) show that game-playing serves as a testbed to enhance general reasoning capabilities (e.g., System 2 reasoning). The current results show a regression in general capabilities. Future work should focus on demonstrating unique advantages of using LLMs in this domain that traditional algorithms cannot achieve.

---

### Decision · Program_Chairs · 2026-01-26

Reject